# THINK IN GRAPHS: INFRASTRUCTURE AND BENCHMARK FOR LARGE LANGUAGE MODEL REASONING FRAMEWORKS

## ABSTRACT

Enhancing the reasoning ability of Large Language Models (LLMs) has become a central focus of current research. While approaches based on prompt engineering have significantly improved LLM performance, the increasing complexity of reasoning frameworks has led to higher development costs. Moreover, these frameworks often require extensive redesigns to actually work on different tasks, with their performance heavily dependent on these specific designs. This creates challenges in establishing clear and consistent evaluation benchmarks. To address these issues, we propose a unified infrastructure that represents reasoning processes as graphs, thereby standardizing and structuring the reasoning workflow. This approach enables more consistent and efficient implementation of diverse reasoning frameworks, facilitates objective comparisons, and supports deeper analysis through graph algorithms. Building on this infrastructure, we develop an LLM reasoning benchmark and demonstrate its effectiveness through multiple experiments, enabling more comprehensive evaluation and analysis. *Code and data can be found in* https://anonymous.4open.science/r/210-5DD5/.

## 1 INTRODUCTION

The enhancement of reasoning ability has become a major focus in current research on LLMs (Huang & Chang, 2023). With the emergence of the Chain-of-Thought (CoT) framework (Wei et al., 2022b), prompt-based reasoning optimization methods have gained widespread applications (Hao et al., 2024). Through CoT, LLMs can reason more transparently and better handle complex tasks. Further advancements, such as Tree of Thoughts (ToT) (Yao et al., 2023b) and Graph of Thoughts (GoT) (Besta et al., 2024b), represent the reasoning process in more complex tree-like or graph-like structures, as well as others, facilitating more sophisticated forms of reasoning (Yao et al., 2024; Shin & Kim, 2025; Sel et al., 2024; Zhou et al., 2023b).

However, as more complex and powerful reasoning frameworks are proposed, implementing them incurs increasingly higher costs, including those related to coding and prompt design (McDonald et al., 2024). Furthermore, these frameworks often require frequent redesigns of prompts and program structures to perform effectively across different tasks (Gao et al., 2025). This not only results in development inefficiencies but also creates a strong dependence between the performance of these methods and their specific designs, making it challenging to establish clear and consistent benchmarks.

In fact, logical reasoning is inherently a highly complex and difficult-to-quantify process (Shojaee et al., 2025). Reasoning involves not only the organization and deduction of information but also the integration of information across multiple dimensions and layers. Different reasoning paths may lead to the same conclusion, and whether the reasoning steps within these paths are considered "reasonable" or "correct" often lacks a unified standard. In the philosophy and cognitive science of human thinking, logical reasoning is often viewed as a field filled with ambiguity and uncertainty (Stenning & Van Lambalgen, 2012). Therefore, despite the significant advancements made by LLMs in prompt-based reasoning optimization, **how to avoid the continuous redevelopment of reasoning frameworks and how to objectively and fairly evaluate these reasoning processes remain unresolved challenges.**

To address the above issue, it is necessary to develop a unified framework that can standardize and structure the reasoning process of LLMs. In fact, if the reasoning steps are represented as nodes and the relationships between these steps are represented as edges, all reasoning processes can be expressed as graphs. This is because all reasoning architectures—whether chains, trees, or other forms—are essentially specialized instances of a graph. Moreover, by representing the reasoning process as structured data in the form of a graph, we can make a more objective comparison of these reasoning processes, e.g., using quantifiable graph distances to measure the differences between reasoning procedures. Additionally, if all reasoning processes can be represented as graphs, a unified graph-based infrastructure would also, in turn, enable the implementation of any reasoning framework. Figure 1 provides an intuitive illustration of such viewpoints.

Building on this perspective, we have constructed an infrastructure for LLM reasoning called *Think in Graphs* (TiG). TiG implements different LLM reasoning frameworks in a unified manner. Specifically, all reasoning processes in TiG are specified by a configuration file.

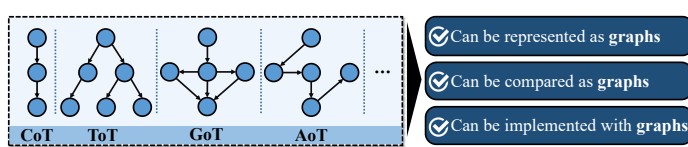

Figure 1: Illustration of different reasoning frameworks.

Based on such a file, TiG continuously generates new thoughts, which are added to the graph-based reasoning flow, enabling the ongoing progression of reasoning. For users of TiG, the only requirement is to define the configuration file, which eliminates the need to rebuild the entire reasoning framework. Additionally, TiG can save the LLM reasoning process in graph form and use this data structure for unified and objective comparisons. Furthermore, graph-based algorithms, such as our proposed graph kernel (introduced later), can be applied to the analysis of logical reasoning, thereby enabling more diverse and in-depth evaluations. Building on TiG and the collected reasoning tasks of various types, we constructed a prompting-based reasoning benchmark for LLMs. We then conducted extensive experiments with different reasoning frameworks on this benchmark to gain deeper insights and to demonstrate the practicality of the proposed TiG.

Our contributions are as follows:

- We design TiG, a unified and efficient infrastructure that facilitates the rapid implementation of diverse LLM reasoning frameworks, serving as a foundation to support ongoing research on prompt engineering for reasoning.

- With TiG, we design a series of new metrics for analyzing LLM reasoning logic, including a novel graph kernel.

- Based on TiG and a dataset with a variety of test tasks, we construct a benchmark for prompt engineering for reasoning.

- We conduct a series of analytical experiments based on the proposed benchmark and derive the corresponding conclusions.

## 2 RELATED WORKS

### 2.1 REASONING WITH PROMPTING

Recent research has increasingly focused on designing logically consistent prompts that improve reasoning performance, enabling LLMs to tackle complex tasks more effectively. The CoT framework (Wei et al., 2022a) has inspired advances like Auto-CoT (Zhang et al., 2023), which automates and optimizes reasoning outputs, and LogiCoT (Zhao et al., 2024), which integrates symbolic logic for refinement. Prompt Sketching (Beurer-Kellner et al., 2024) and CCoT (Mitra et al., 2024) improve control over reasoning, while RASC (Wan et al., 2025) boosts consistency and reduces sampling costs. More complex structures like ToT (Yao et al., 2023b), GoT (Besta et al., 2024a), and GoTR (Yao et al., 2024) enhance multi-step and multimodal reasoning. EGoT (Shin & Kim, 2025) optimizes inference paths, and ThoT (Zhou et al., 2023b) and AoT (Sel et al., 2024) structure reasoning hierarchically and algorithmically, enabling more efficient exploration of complex reasoning paths.

## 2.2 BENCHMARKING LLM REASONING

Recent research on the reasoning abilities of LLMs has led to the development of several benchmarks. MMLU (Hendrycks et al., 2021), BIG-bench (Srivastava et al., 2022), and HELM (Liang et al., 2022) provide comprehensive multi-task evaluations. MT-Bench (Zheng et al., 2023) focuses on multi-turn dialogue reasoning, while OpenAI Evals (OpenAI, 2023b) facilitates benchmark sharing. Specific datasets for reasoning include BBH (Suzgun et al., 2022), GSM8K (Cobbe et al., 2021), MATH (Hendrycks et al., 2021), ARC (Clark et al., 2018), and DROP (Dua et al., 2019). ReClor (Yu et al., 2020) evaluates logical reasoning, and MME-CoT (Jiang et al., 2025) focuses on CoT reasoning in LLMs. REVEAL (Greyling, 2024) verifies CoT correctness. We approach the problem from a different perspective, constructing a new, more unified thinking infrastructure based on graphs, then benchmarking LLM reasoning with it.

## 3 INFRASTRUCTURE

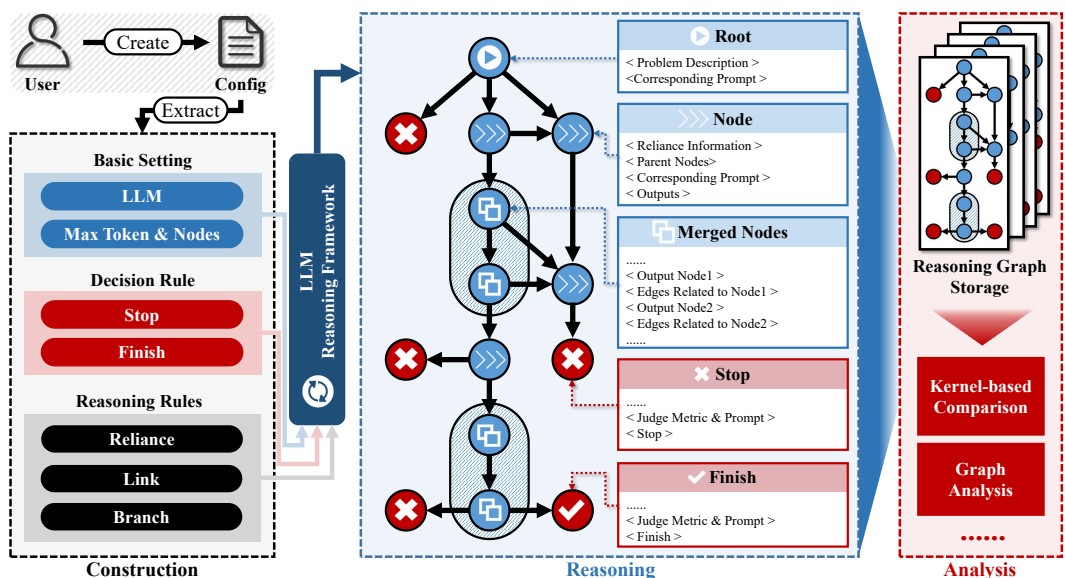

Figure 2: Architecture of the proposed TiG infrastructure.

The overall architecture of TiG is illustrated in Figure 2. In TiG, the entire reasoning process is represented as a directed acyclic graph (DAG), denoted as $G^{(t)} = \{\mathcal{V}^{(t)}, \mathcal{E}^{(t)}\}$, where $t$ indicates the number of reasoning iterations. $\mathcal{V}^{(t)}$ represents the set of nodes in $G^{(t)}$, with each node corresponding to a reasoning step generated by the LLM. Each node effectively encodes a segment of thought. $\mathcal{E}^{(t)}$ denotes the set of edges, where each edge represents a dependency between nodes. The use of the DAG is motivated by its clear causal structure and computational simplicity (Spirtes et al., 2000). Operations such as backtracking (Besta et al., 2024a), which might otherwise introduce cycles, are also represented within acyclic structures. Specifically, a newly generated node resulting from a backtracking operation can be formalized as a common child of the node requiring backtracking and the node it backtracks to. In other words, as the reasoning process unfolds, $G^{(t)}$ is guaranteed to always remain a DAG.

Clearly, the evolution of $G^{(t)}$ with increasing $t$ reflects the ongoing reasoning process of the LLM. TiG constrains and guides this evolutionary process, while also enabling the analysis of $G^{(t)}$. Specifically, the workflow of TiG consists of three phases: construction, reasoning, and analysis. (1) In the **construction phase**, the user provides a configuration file to define the intended reasoning process, including three sets of constraint rules that specify the reasoning behavior within the framework. (2) In the **reasoning phase**, the LLM leverages our framework to extract the defined rules from the user's configuration and execute the reasoning accordingly. (3) In the **analysis phase**, we collect the final $G^{(t)}$ graph and perform result analysis. These phases will be introduced separately below.

## 3.1 CONSTRUCTION PHASE

In this phase, the user constructs the configuration file, specifying the basic settings, the decision rule, and the reasoning rules. The approaches for building each of these three components will be introduced in the following subsections. The implementation details of the configuration file can be found in **Appendix C**, and **Section 3.4** provides a running example.

**Basic settings.** The basic settings include the LLM to be used, as well as the specified limits on the number of tokens and nodes consumed for task execution.

**Decision rule.** This rule specifies how to determine the next action for a given node $v^{(t)} \in G^{(t)}$. To be concise, we omit the superscript $(t)$ for $v^{(t)}$, because the node $v$ itself does not change over time. The decision rule constrains whether the reasoning process should (1) terminate at $v$, (2) treat $v$ as the final answer, or (3) continue reasoning based on $v$. The decision rule is expressed as a textual description.

**Reasoning rules.** These rules apply to nodes that require further reasoning, and they specify the structure and attributes of the subgraph $G^{\text{sub}}_{(v)}$ to be generated. In essence, $G^{\text{sub}}_{(v)}$ represents the newly generated thoughts together with their relationships to the preceding ones. Specifically, each node $v$ corresponds to one reasoning rule, which is selected based on the specific characteristics of $v$, including its position in the graph, its distance from the root node, its attributes, and other contextual features. For the defined set of rules $\Phi = \{\phi_i\}_{i=1}^m$, our infrastructure builds the following mapping:

$$i = s(v, G^{(t-1)}, \Phi), \quad i \in \{1, 2, ..., m\}, \tag{1}$$

where $i$ is the index of the specific rule, $s(\cdot)$ denotes the selection function. $s(\cdot)$ is defined by the user with the configuration file, with details in **Appendix C**.

The selected rule determines, based on the features of node $v$ and the current $G^{(t)}$, all possible child nodes of $v$, their respective parent nodes, the connections between these nodes, and the prompt used for generation. Formally, the child node set $\text{Ch}(v)$ of $v$ can be represented as:

$$\text{Ch}(v) = \{u \in \mathcal{V}^{(t+1)} \mid (v, u) \in \mathcal{E}^{(t+1)}\}, \tag{2}$$

where $\mathcal{E}^{(t+1)}$ and $\mathcal{V}^{(t+1)}$ are generated by applying the rules. The set of parent nodes is:

$$\bigcup_{u \in \text{Ch}(v)} \text{Pa}(u) = \left\{ w \in \mathcal{V}^{(t)} \mid \exists u \in \mathcal{V}^{(t+1)}, (v, u) \in \mathcal{E}^{(t+1)} \wedge (w, u) \in \mathcal{E}^{(t+1)} \right\}. \tag{3}$$

$\text{Pa}(\cdot)$ denotes the parent nodes. The generated graph substructure corresponding to $v$ is:

$$G^{\text{sub}}_{(v)} = \left\{ \text{Ch}(v), \left\{ (u, w) \in \mathcal{E}^{(t+1)} \mid u \in \text{Ch}(v), w \in \text{Pa}(u) \right\} \cap \left\{ (v, u) \in \mathcal{E}^{(t+1)} \mid u \in \text{Ch}(v) \right\} \right\}. \tag{4}$$

In the reasoning process, some methods generate only a single reasoning step—i.e., a single node in $G^{(t)}$—per generation round of LLM, while others may generate multiple reasoning steps at once. Our framework is designed to support both approaches. Specifically, when multiple nodes are generated in a single round, we treat the entire output as a single node during initial processing, and then split it into individual nodes based on predefined delimiters embedded in the infrastructure.

## 3.2 REASONING PHASE

In this phase, the framework processes the nodes and continuously updates the graph until either a final result is obtained or the maximum token (or node) limit for reasoning is reached. At the $t$-th update step, only the nodes newly generated in step $(t-1)$, i.e., $u \in G^{(t-1)} \setminus G^{(t-2)}$, are selected. This is because all other nodes are either already terminated or are ancestor nodes of the newly generated nodes, and thus no longer represent active reasoning processes. Excluding them helps reduce computational cost. The detailed reasoning procedure is provided in Algorithm 1.

---

**Algorithm 1** Reasoning Procedure of TiG

---

**Require:** Problem description, decision rule, evolution rule set $\Phi$
**Ensure:** Final answer to the problem
1: Initialize graph $G^{(0)}$ with a single root node based on the problem description
2: Set $t \leftarrow 1$
3: **while** Total token usage or node count has not exceeded the maximum limit **do**
4:      Select newly generated nodes: $\mathcal{X} \leftarrow G^{(t-1)} \setminus G^{(t-2)}$
5:      **for all** $v \in \mathcal{X}$ **do**
6:          Apply decision rule to determine whether $v$ should be an answer node or terminated
7:          **if** $v$ is an answer node **then**
8:              Output the answer and terminate the reasoning process
9:          **else if** reasoning based on $v$ should be stopped **then**
10:            Skip to the next node
11:          **else**
12:              Identify evolution rule index: $i \leftarrow s(v, G^{(t-1)}, \Phi)$
13:              Retrieve $\phi_i$ from $\Phi$ and the structure of $G_{(v)}^{\text{sub}}$
14:              Generate node features of $G_{(v)}^{\text{sub}}$ via LLM using $v$, $G^{(t-1)}$, and $\phi_i$.
15:              Integrate $G_{(v)}^{\text{sub}}$ into $G^{(t)}$
16:          **end if**
17:      **end for**
18:      Update round index: $t \leftarrow t + 1$
19: **end while**

---

### 3.3 ANALYSIS PHASE

We represent the entire reasoning process using the graph obtained at the final time step, denoted as $G$. This graph serves as a compact and interpretable abstraction of the sequence of intermediate reasoning steps. Based on $G$, we are able to perform more precise and fine-grained analyses. In particular, we can extract the exact set of reasoning paths that contribute to the final answer—namely, the union of all directed paths from the root node $r$ to the answer node $a$, denoted by $\mathcal{P}_{r \to a}$. This allows us to compute the proportion of nodes involved in generating the final answer as:

$$\lambda = \frac{|\mathcal{P}_{r \to a}|}{|\mathcal{V}|}.$$

Furthermore, the structural properties of $G$ enable us to identify and quantify redundant nodes, calculate the proportion of redundant tokens, and track the precise number of instances when the reasoning reaches a dead end. These metrics are thoroughly analyzed in the experimental section.

Given this graph-based representation of the LLM's reasoning trajectory, we further introduce the concept of a graph kernel to formally measure the similarity between different reasoning processes. Specifically, we propose an extension to the traditional Weisfeiler-Lehman (WL) kernel (Shervashidze et al., 2011) that incorporates the directionality of $G$ and the answer-contributing ratio $\lambda$. The resulting kernel, termed the *Reasoning Graph Weisfeiler-Lehman (RGWL)* kernel, is defined as follows:

$$\mathbb{K}_{\text{RGWL}}^{(h)}(G, G') = \sum_{i=1}^{h} \Big( \Big\langle \eta \Big( \widetilde{\psi}^{(i)} \left( \rho \left( \tau \left( G \right) \right) \right) \Big), \eta \Big( \widetilde{\psi}^{(i)} \left( \rho \left( \tau \left( G' \right) \right) \right) \Big) \Big\rangle$$
$$+ \lambda \lambda' \Big\langle \eta \Big( \widetilde{\psi}^{(i)} \left( \rho \left( G \right) \right) \Big), \eta \Big( \widetilde{\psi}^{(i)} \left( \rho \left( G' \right) \right) \Big) \Big\rangle \Big), \tag{5}$$

where $\tau(\cdot)$ extracts the answer-contributing subgraph $\mathcal{P}_{r \to a}$, and $\rho(\cdot)$ performs KNN-based node labeling by clustering node features from both $G$ and $G'$ and assigning labels to each cluster. $\widetilde{\psi}^{(i)}(\cdot)$ denotes the $i$-th round of label propagation as in the WL kernel, but restricted to the direction of edges. $\lambda'$ is the answer-contributing ratio of $G'$. The function $\eta(\cdot)$ computes a histogram of node labels after each round. Further details are provided in **Appendix** E.

We further prove that the RGWL kernel is positive semi-definite, ensuring that it defines a valid inner product in a Reproducing Kernel Hilbert Space (RKHS). This theoretical guarantee supports

the validity of subsequent analyses and allows the RGWL kernel to be broadly employed in a variety of kernel-based learning algorithms.

**Proposition 1.** *The kernel matrix $K_{RGWL}^{(h)}$ defined by $\mathbb{K}_{RGWL}^{(h)}(\cdot, \cdot)$ is positive semi-definite.*

The formal proof of this proposition is provided in **Appendix D**.

## 3.4 RUNNING EXAMPLE

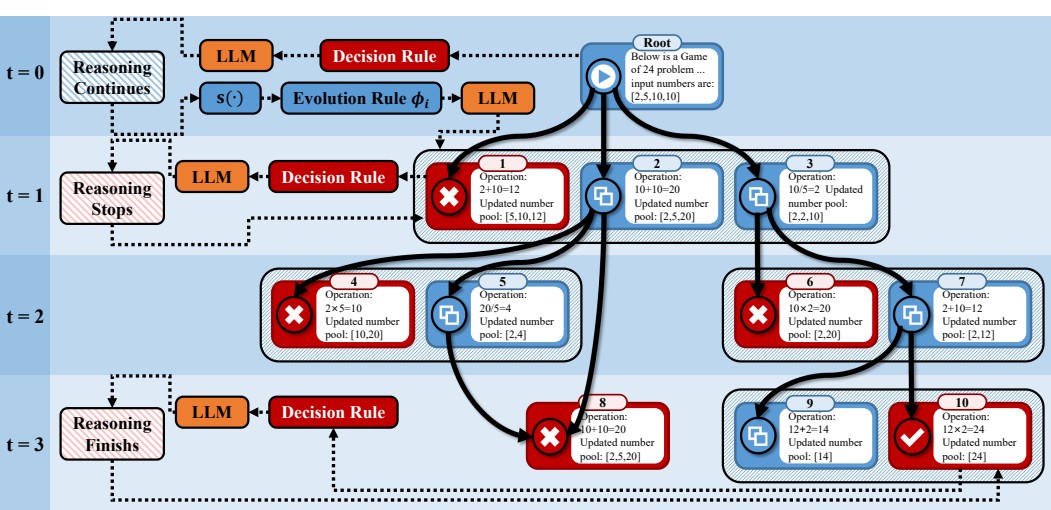

Figure 3: A running example on Game of 24.

Here, we use a practical example to demonstrate the infrastructure. The example is a ToT (Yao et al., 2023a) framework specifically built for the Game of 24 problem with TiG.

The first step is to construct the configuration file that specifies the decision rules and reasoning rules. The specific content of the configuration file used in this example is provided in **Appendix C**. Within it, the decision rule states that if the answer can be calculated to 24, the result should be returned. If neither the current node nor any historical nodes can lead to 24, the reasoning based on the current node should be terminated. The reasoning rules state that: If the node is the root, then generate three child nodes for further reasoning. If the current node cannot lead to 24, but its parent node can, then backtrack and generate two child nodes based on that parent node. In all other cases, generate two child nodes for further reasoning. Additionally, the reasoning rules establish that, regardless of the number of nodes generated, the generation of each node's child nodes is carried out through a single interaction with the LLM.

Next, reasoning is carried out based on the configuration file. The entire reasoning process is illustrated in Figure 3. Initially, at time $t = 0$, reasoning begins at the root node. The root node essentially serves as a description of the problem. As shown in the figure, the content of the root node is evaluated by the decision rule and the LLM to determine whether reasoning should continue. Once reasoning is confirmed to proceed, the corresponding rule from the reasoning rules is selected. Since this is the root node, the rule specifically associated with the root is applied, leading to the generation of three child nodes.

At $t = 1$, each newly generated node is evaluated for whether reasoning should continue. As illustrated in the figure, the reasoning process terminates on Node 1 after applying the decision rule. In contrast, two subsequent reasoning nodes are generated based on Node 3. At $t = 2$, Node 5, which matches the backtracking rule, reinitiates the reasoning process together with its parent node. Specifically, Node 5 generates a subgraph $G_{(5)}^{\text{sub}}$, which includes Nodes 5, 2, and 8. The edges of the subgraph are represented by the pairs $(5, 8)$ and $(2, 8)$. Ultimately, Node 10 meets the necessary criteria and produces the final answer.

The resulting graph, along with the associated attributes, can subsequently be utilized for kernel-based graph analysis.

# 4 BENCHMARK

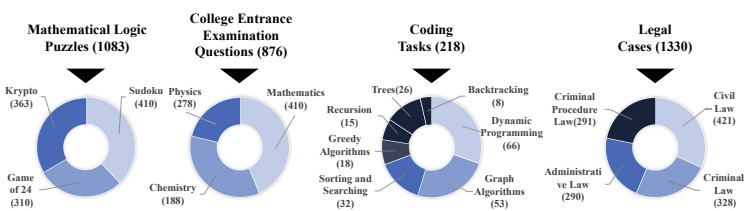

Figure 4: Category and subcategory distribution of questions in TiG Benchmark.

Table 1: Key statistics of the TiG Benchmark

| Statistic | Number |
|---|---|
| Total questions | 3529 |
| Total categories | 4 |
| Total subcategories | 27 |
| Answer in text form | 2313 |
| Answer in function form | 1207 |
| Total implemented methods | 55 |
| Implemented methods per category | 13.75 |

Based on the proposed TiG, we construct the corresponding TiG Benchmark to analyze prompt-engineering-based LLM reasoning. For test data, we have collected a diverse set of reasoning tasks with varying complexity, ensuring that the evaluation results can be reliably validated. Furthermore, we have implemented different reasoning frameworks for each of these tasks using our proposed infrastructure. Specifically, we collected a total of four categories of data, namely Mathematical Logic Puzzles, College Entrance Examination Questions, Coding Tasks, and Legal Cases. The detailed composition of these datasets is illustrated in Figure 4. Among them, the problems in Mathematical Logic Puzzles were constructed based on three different types of mathematical games. The problems in the College Entrance Examination Questions were derived from China's national college entrance examination. The Coding Tasks were collected from various online sources containing programming problems. As for Legal Cases, they consist of publicly released court cases, where the LLM is required to provide appropriate decisions based on the case descriptions. We provide details concerning the collection procedure along with the provided data.

Answers for the questions come in two formats: textual and functional. Textual answers consist of written content, such as selections for multiple-choice questions, authoritative judicial rulings from official institutions, and other text-based responses. The functional answers, on the other hand, are correctness-checking functions specifically designed for certain problems, e.g., a validation function to check whether a submitted arithmetic expression evaluates to 24. Furthermore, for the questions, our benchmark implements five state-of-the-art reasoning frameworks: CoT (Wei et al., 2022a), ToT (Yao et al., 2023a), GoT (Besta et al., 2023), AoT (Sel et al., 2024), and EGoT (Shin & Kim, 2025). Table 1 presents the specific statistics related to the benchmark.

# 5 EXPERIMENTS

## 5.1 STATISTICAL EVALUATIONS

Table 2: Results on Mathematical Logic Puzzles. The best performance is highlighted in **bold**, and the second-best is indicated with underline.

| Method | Accuracy | Time Cost (s) | Node Redundancy | Thought Redundancy | Count of Invalid Branches | Root-Answer Shortest Path Length |
|---|---|---|---|---|---|---|
| CoT | 62.08 ± 2.11 | **26.87** | **0.00 ± 0.00** | **0.00 ± 0.00** | **0.05 ± 1.51** | 4.68 ± 1.05 |
| ToT | 85.37 ± 7.03 | 78.16 | 71.96 ± 3.40 | 73.01 ± 6.42 | 6.10 ± 0.68 | 6.88 ± 0.03 |
| AoT | 89.55 ± 2.22 | 37.42 | 60.15 ± 5.04 | 63.31 ± 8.23 | 3.06 ± 0.08 | **3.03 ± 0.56** |
| GoT | 87.85 ± 8.47 | 74.65 | 79.01 ± 4.42 | 80.31 ± 6.03 | 5.47 ± 1.10 | 5.45 ± 0.98 |
| EGoT | **89.97 ± 7.67** | 70.78 | 51.10 ± 3.01 | 48.42 ± 7.04 | 4.99 ± 0.72 | 5.01 ± 1.43 |

Based on the constructed TiG Benchmark, we conducted a series of analytical experiments on the implemented reasoning frameworks mentioned above, i.e., CoT, ToT, GoT, AoT, and EGoT, in order to validate the usability of our infrastructure and further investigate the characteristics of these reasoning paradigms. We first performed statistical evaluations across multiple metrics, including accuracy, redundancy, and the number of invalid branches.

Specifically, based on our graph representation of reasoning logic, we propose the following new metrics: node redundancy, thought redundancy, count of invalid branches, and root–answer shortest

Table 3: Results on College Entrance Examination Questions. The best performance is highlighted in **bold**, and the second-best is indicated with underline.

| Method | Accuracy | Time Cost (s) | Node Redundancy | Thought Redundancy | Count of Invalid Branches | Root-Answer Shortest Path Length |
|--------|----------|---------------|-----------------|--------------------|--------------------------|----------------------------------|
| CoT | 48.06 ± 1.25 | 59.74 | **0.00 ± 0.00** | **0.00 ± 0.00** | **0.17 ± 0.06** | **3.57 ± 1.46** |
| ToT | 62.42 ± 0.96 | 154.33 | 78.41 ± 4.10 | 74.60 ± 4.16 | 5.08 ± 0.94 | 5.63 ± 1.01 |
| AoT | 65.17 ± 2.11 | **65.96** | 53.21 ± 4.13 | 51.56 ± 4.10 | 4.10 ± 0.10 | 4.52 ± 1.34 |
| GoT | 67.22 ± 3.31 | 101.23 | 69.33 ± 3.31 | 67.67 ± 4.30 | 5.96 ± 1.32 | 5.17 ± 0.21 |
| EGoT | **69.89 ± 1.30** | 135.47 | 50.51 ± 6.22 | 50.62 ± 6.10 | 5.03 ± 1.58 | 4.06 ± 0.85 |

Table 4: Results on Coding Tasks. The best performance is highlighted in **bold**, and the second-best is indicated with underline.

| Method | Accuracy | Time Cost (s) | Node Redundancy | Thought Redundancy | Count of Invalid Branches | Root-Answer Shortest Path Length |
|--------|----------|---------------|-----------------|--------------------|--------------------------|----------------------------------|
| CoT | 48.23 ± 7.23 | **95.44** | **0.00 ± 0.00** | **0.00 ± 0.00** | **0.12 ± 0.08** | **3.56 ± 0.01** |
| ToT | **72.56 ± 9.21** | 143.92 | 74.96 ± 5.15 | 78.47 ± 6.01 | 3.56 ± 1.11 | 4.10 ± 0.75 |
| AoT | 82.12 ± 1.03 | 136.48 | 52.12 ± 3.08 | 56.34 ± 2.45 | 3.93 ± 1.23 | 5.22 ± 1.33 |
| GoT | 72.45 ± 2.56 | 140.01 | 72.56 ± 7.12 | 75.31 ± 9.04 | 3.12 ± 1.14 | 5.89 ± 0.31 |
| EGoT | 82.68 ± 9.32 | 131.45 | 69.01 ± 4.25 | 71.44 ± 4.56 | 4.15 ± 1.35 | 4.68 ± 1.10 |

Table 5: Results on Legal Cases. The best performance is highlighted in **bold**, and the second-best is indicated with underline.

| Method | Accuracy | Time Cost (s) | Node Redundancy | Thought Redundancy | Count of Invalid Branches | Root-Answer Shortest Path Length |
|--------|----------|---------------|-----------------|--------------------|--------------------------|----------------------------------|
| CoT | **43.33 ± 2.05** | **70.74** | **0.00 ± 0.00** | **0.00 ± 0.00** | **0.21 ± 0.06** | **3.57 ± 1.46** |
| ToT | 58.33 ± 1.02 | 154.33 | 71.41 ± 4.10 | 78.60 ± 4.16 | 5.08 ± 0.94 | 5.63 ± 1.01 |
| AoT | 61.67 ± 2.12 | 113.96 | 65.21 ± 4.13 | 65.56 ± 4.10 | 4.10 ± 0.10 | 4.52 ± 1.34 |
| GoT | 63.83 ± 3.31 | 135.47 | 76.33 ± 3.31 | 80.67 ± 3.40 | 4.96 ± 1.32 | 5.17 ± 0.21 |
| EGoT | 65.01 ± 1.30 | 129.23 | 65.51 ± 6.22 | 70.62 ± 6.10 | 6.23 ± 1.58 | 4.16 ± 0.85 |

path length. Node redundancy and thought redundancy respectively represent the percentage of redundant nodes and tokens relative to the total numbers. Redundant nodes and tokens are defined as those not lying on any path from the root node to the answer node, as well as the tokens contained within such nodes. The count of invalid branches measures the number of terminated reasoning branch that fail to acquire the answer. Root–answer shortest path length is the shortest distance from the root node to the answer node in the graph.

The experimental results are presented in Tables 2–5. As shown in the tables, the four newly introduced metrics enable a more in-depth analysis and comparison of the reasoning process. Based on these metrics, we observe that more complex reasoning frameworks tend to include a larger number of redundant nodes; however, the actual path length from the question to the answer does not vary significantly. When considered alongside the accuracy of different methods, this suggests that complex frameworks explore a wider range of possibilities in order to achieve higher accuracy. Figure 5 summarizes the average performance of each framework across different datasets. From the figure, it can be observed that the legal dataset is the most challenging one, while the coding dataset exhibits the largest performance gap among models.

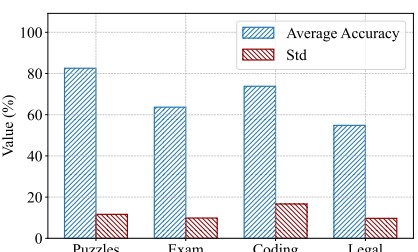

Figure 5: Average accuracy and standard deviation across datasets.

## 5.2 DEEPER INSIGHTS

Additionally, we performed a similarity analysis between different reasoning approaches based on the RGWL kernel. First, we compared the similarity between the reasoning processes generated by a single reasoning framework, GoT, across different problems. The visualization of the comparison results is shown in Figure 6. It can be observed that for the coding and legal datasets, the consistency of reasoning between correct answers is higher, as indicated by higher RGWL kernel output. In

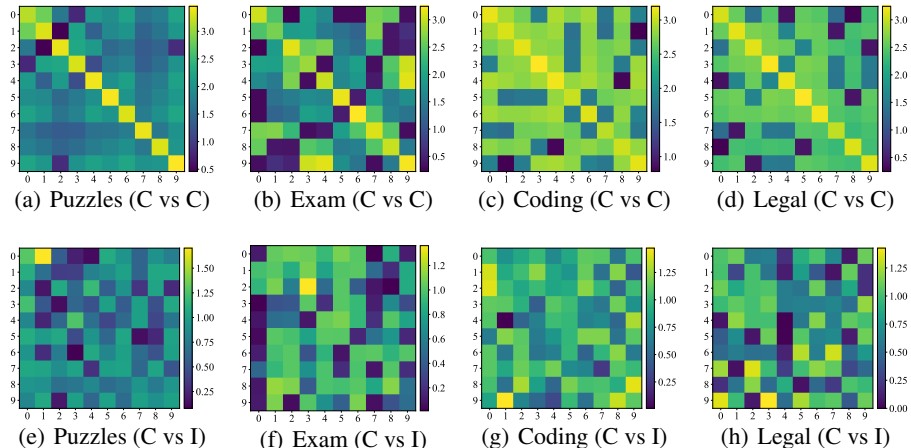

Figure 6: Visualization of RGWL results with GoT: "C vs C" compares ten reasoning graphs with correct answers, while "C vs I" compares ten correct-answer graphs with ten incorrect ones. Brighter colors indicate higher RGWL output and greater similarity.

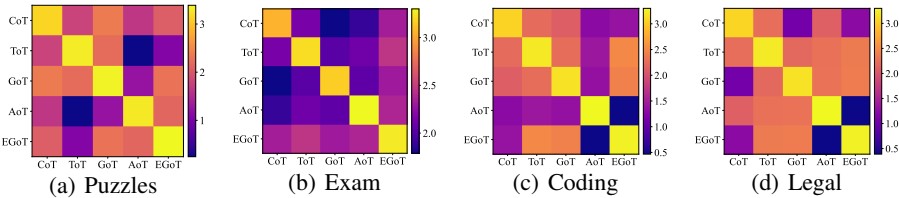

Figure 7: Visualization of RGWL computation results for different reasoning processes with different reasoning frameworks.

contrast, the distance between incorrect reasoning and correct reasoning is significantly greater than the distance between two correct reasoning processes.

Furthermore, we compared the similarity between the reasoning processes generated by different frameworks across various problems, with the results shown in Figure 7. It can be observed that the similarity between reasoning processes produced by different frameworks is highest in the legal dataset, which is consistent with our previous experimental findings. This observation indicates that the knowledge structures and decision rules in the legal domain are relatively stable and standardized, leaving less room for divergent reasoning paths. As a result, even when different reasoning frameworks are applied, they tend to converge on similar reasoning trajectories and final conclusions.

In contrast, datasets such as coding tasks or mathematical puzzles allow multiple solution strategies and a more open-ended reasoning space, leading to lower similarity between frameworks. This demonstrates that our graph-based analysis captures not only outcome accuracy but also the structural consistency and diversity of reasoning processes.

## 6 CONCLUSION

In this paper, we propose the TiG infrastructure based on graph structures, which enables the construction of diverse prompt-engineering-based LLM reasoning frameworks in a unified and streamlined manner. Building upon TiG, we introduce a broader set of reasoning logic evaluation metrics and develop a benchmark for comparing different reasoning frameworks, on which we conduct a series of experiments and analyses.

## REPRODUCIBILITY STATEMENT

Our theoretical results have been rigorously proven, and the corresponding proofs are provided in Appendix D. Additionally, our experiments provide both data and code to ensure reproducibility. These resources are included in the anonymous link https://anonymous.4open.science/r/210-5DD5/, with further details available in the accompanying README.md file.

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

## A  USAGE OF LARGE LANGUAGE MODEL

In our paper, we used LLMs to assist with polishing the writing, including correcting grammatical errors and making the sentences more consistent with academic English writing conventions.

## B  EXTENDED RELATED WORKS

### B.1  LARGE LANGUAGE MODELS

Since the introduction of the Transformer architecture (Vaswani et al., 2017), LLMs have undergone rapid and transformative development. The Transformer, with its self-attention mechanism and strong parallelism capabilities, quickly became the foundational architecture for modern neural language models, setting a universal paradigm for subsequent designs.

Based on this architecture, OpenAI released the initial versions of the GPT series in 2018 (Radford, 2018), leveraging autoregressive language modeling to achieve notable performance in text generation. This was followed by GPT-2 (Radford et al., 2019), which significantly expanded model capacity and demonstrated strong generalization across diverse tasks. GPT-3, introduced in 2020 (Brown et al., 2020), scaled to 175 billion parameters and marked a major leap in LLM capabilities. GPT-3.5 further optimized inference efficiency and improved contextual understanding, especially in dialogue-oriented tasks. GPT-4 (OpenAI, 2023a) introduced substantial advances in logical reasoning, knowledge integration, and alignment with human values, enabling its application to more complex and multi-modal tasks. The most recent iteration, GPT-4o, focuses on enhancing safety, robustness, and ethical alignment, making it particularly well-suited for high-stakes decision-making scenarios.

In parallel with the GPT lineage, BERT (Devlin et al., 2019) was introduced in 2019, pioneering bidirectional contextual learning through masked language modeling (MLM), which marked a departure from the limitations of unidirectional models. Building on BERT, numerous improved variants have been proposed: RoBERTa (Liu et al., 2019) removed the next sentence prediction objective and used more extensive pretraining data; ALBERT (Lan et al., 2020) introduced parameter sharing and factorized embedding layers to reduce redundancy; BERTweet (Nguyen et al., 2020) targeted social media text processing; and BigBird (Zaheer et al., 2020) employed sparse attention mechanisms to effectively handle longer input sequences.

More recently, a wave of novel LLMs has emerged, further diversifying the research landscape. Meta's LLaMA series (Touvron et al., 2023; Rozière et al., 2023; AI, 2024) has contributed significantly to the development of efficient, open-source, and lightweight models. Subsequently, LLaMA 4 was introduced as an open-source large language model series featuring a Mixture-of-Experts architecture, native multimodal capabilities, industry-leading extended context length, and enhanced multilingual support, achieving significant breakthroughs in both performance and efficiency (Meta, 2025). Meanwhile, Anthropic's Claude series (Anthropic, 2023b;a; 2024) emphasizes model alignment, safety, and controllability, proposing new mechanisms for responsible AI deployment. In addition, several other notable models have been introduced in recent years, including Google's Gemini (Anil et al., 2023), Alibaba's Tongyi Qianwen (Yang et al., 2024), Baidu's ERNIE (Sun et al., 2021), Amazon Nova(AGI et al., 2025), Mistral (Jiang et al., 2023), Falcon (Penedo et al., 2023), and PaLM (Chowdhery et al., 2023). Each of these models introduces distinct innovations in architectural design, code generation, multilingual capability, training efficiency, or open-access availability, collectively advancing the capabilities and diversity of the LLM ecosystem.

### B.2  PROMPT ENGINEERING

With the widespread deployment of LLMs, their capabilities in natural language understanding and generation have continued to surpass expectations. However, effectively guiding these models to produce accurate, logically coherent, and structurally consistent outputs remains a key challenge. Prompt Engineering has emerged as a critical solution to this issue and has rapidly evolved in recent years, forming a systematic framework encompassing strategies such as reasoning enhancement, hallucination mitigation, structured task adaptation, and interactive optimization.

To improve the reasoning capabilities of LLMs, researchers initially proposed the Chain of Thought (CoT) approach, which significantly enhances the model's deductive reasoning in complex tasks—such as mathematical problem solving and textual inference—by prompting it to generate explicit intermediate reasoning steps (Wei et al., 2022a). Building upon this foundation, methods such as Program of Thoughts (PoT) and Structured Chain-of-Thought (SCoT) further modularize the reasoning process, making them particularly effective for code generation, logic programming, and multi-stage computation tasks (Chen et al., 2023; Li et al., 2023b). Additionally, techniques like Flow Engineering have been proposed to improve the semantic consistency and execution fidelity of code-related prompts, thereby expanding the design space for structured prompt generation (Ridnik et al., 2024).

Hallucination mitigation constitutes another central focus of prompt engineering. Classical approaches such as Retrieval-Augmented Generation (RAG) integrate external knowledge sources into the generation process, providing factual grounding and improving the factual accuracy of model outputs from the outset (Lewis et al., 2020). Meanwhile, post-hoc verification methods—such as Chain-of-Verification (CoVe), Chain-of-Note (CoN), and Chain-of-Knowledge (CoK)—introduce layered review mechanisms, filtering out false or inconsistent content through multi-stage validation, citation checking, and cross-examination (Dhuliawala et al., 2024; Yu et al., 2023).

As LLMs are increasingly deployed in open-domain environments, enhancing their interactivity and understanding of user intent has become a key extension of prompt engineering. Interactive Question Answering (Interactive QA) frameworks allow models to obtain real-time feedback through multi-turn dialogue, enabling dynamic adjustment of responses (Yao et al., 2023c; Masson et al., 2024). Concurrently, research has focused on the automation and personalization of prompt selection processes—for example, through techniques that match prompt templates to task-specific contexts (Zhou et al., 2023a)—as well as on modeling user intent for tasks involving emotion control and stylistic adaptation (Diao et al., 2024; Li et al., 2023a).

Currently, prompt engineering is undergoing a transition from empirically driven practices to a more theory-guided scientific paradigm. Scholars have proposed systematic frameworks that integrate diverse prompting techniques (Schulhoff et al., 2025), while also exploring human-in-the-loop methodologies to enable more controllable and robust generation systems (Shah, 2025). Moreover, emerging methods such as Self-Rewarding Language Models and LLM-Augmented LLMs aim to build prompt learning systems that possess self-evaluation and cooperative expansion capabilities, signaling a broader shift toward modular, self-optimizing prompt engineering paradigms (Yuan et al., 2025; Bansal et al., 2024).

### B.3 REASONING WITHIN PROMPTING

Recent research has increasingly focused on designing logically consistent prompts that improve reasoning performance, enabling LLMs to tackle complex tasks more effectively. Following the Chain-of-Thought (CoT) framework (Wei et al., 2022a), Auto-CoT (Zhang et al., 2023) introduces an automated pipeline that samples diverse problems and utilizes zero-shot CoT outputs, followed by post-processing to filter and optimize reasoning chains. LogiCoT (Zhao et al., 2024) integrates the principle of Reductio ad Absurdum from symbolic logic to iteratively verify and correct the reasoning process, thereby reinforcing logical rigor.

In parallel, Prompt Sketching (Beurer-Kellner et al., 2024) proposes a structured prompt template to steer the model's reasoning within a predefined format, achieving more controllable logical pathways. Compositional Chain-of-Thought (CCoT) (Mitra et al., 2024) further extends the CoT framework to multi-modal scenarios, promoting cross-modal reasoning. Other studies (Feng et al., 2023) have also explored the theoretical and empirical performance of CoT on mathematical reasoning tasks. To improve output robustness, the Reasoning-Aware Self-Consistency (RASC) framework (Wan et al., 2025) augments traditional self-consistency mechanisms with a dynamic evaluation of the coherence between each reasoning trace and its final answer. By integrating score-driven stopping strategies and weighted voting, RASC not only reduces sampling cost by approximately 70% but also improves predictive accuracy and reasoning fidelity.

Beyond linear reasoning chains, recent efforts have explored more complex topological representations of thought to enhance LLMs' logical modeling capabilities. The Tree of Thoughts (ToT) (Yao et al., 2023b) introduces a branching structure that enables models to search and evaluate multiple

reasoning paths. The Graph of Thoughts (GoT) (Besta et al., 2024a) models reasoning as a graph-based process, well-suited for complex, multi-step, and multi-source reasoning tasks. Similarly, Graph-of-Thought Reasoning (GoTR) (Yao et al., 2024) explicitly encodes inter-node relationships within reasoning paths, particularly effective for integrating heterogeneous information in multi-modal contexts. Building on these, Enhancing Graph of Thoughts (EGoT) (Shin & Kim, 2025) further optimizes the design of inference paths within graph structures, enhancing both reasoning efficiency and consistency for complex tasks.

Additionally, several studies have proposed structurally organized reasoning paradigms. Thread of Thought (ThoT) (Zhou et al., 2023b) advocates decomposing complex problems into hierarchical and sequential "threads of thought" to facilitate more systematic reasoning. Meanwhile, Algorithm of Thoughts (AoT) (Sel et al., 2024) embeds algorithmic reasoning structures into the prompt context, guiding the model to emulate procedural execution. This strategy leverages the recursive dynamics of LLMs, enabling the construction and exploration of sophisticated reasoning paths with minimal interaction.

### B.4 Benchmarking LLM Reasoning

In recent years, research on the reasoning abilities of LLMs has been growing rapidly. As a result, evaluating the reasoning capabilities of LLMs has become a hot topic in research. MMLU (Hendrycks et al., 2021) provides a large-scale multi-task language understanding benchmark that covers tasks from 57 different domains. Similarly, BIG-bench (Srivastava et al., 2022) includes over 204 diverse tasks aimed at comprehensively testing the model's abilities. HELM (Liang et al., 2022) emphasizes a holistic perspective on model evaluation, looking at both the scenarios and the metrics to gain a thorough understanding of a model's capabilities. MT-Bench (Zheng et al., 2023) is a multi-turn dialogue reasoning benchmark that focuses on the reasoning abilities in multi-turn dialogue scenarios. OpenAI Evals (OpenAI, 2023b) is a general evaluation framework designed to facilitate the development and sharing of evaluation benchmarks for LLMs by the community. For reasoning evaluation, BBH (Suzgun et al., 2022) is a set of the 23 most challenging sub-tasks from BIG-bench, specifically assessing models' performance on complex reasoning tasks. GSM8K (Cobbe et al., 2021) is a dataset containing 8,500 elementary school math application problems, used to evaluate the model's mathematical reasoning abilities. The MATH (Hendrycks et al., 2021) dataset includes 12,500 high school math competition questions, primarily testing the model's reasoning abilities on advanced math problems. ARC (Clark et al., 2018) is a dataset designed to evaluate LLMs' abilities in scientific question answering and common-sense reasoning. DROP (Dua et al., 2019) is a reading comprehension dataset used to assess LLMs' abilities in discrete reasoning. ReClor (Yu et al., 2020) focuses on evaluating LLMs' abilities in logical reasoning. MME-CoT (Jiang et al., 2025) is a benchmark specifically designed to evaluate CoT reasoning performance in large multimodal models (LMMs). REVEAL (Greyling, 2024) is a benchmark for verifying the correctness of CoT reasoning chains.

## C Details Concerning Construction Phase

As described in the main text, in our framework, users only need to prepare a single configuration file to complete the relevant setup. The structure and content of this configuration file are detailed below.

The configuration file primarily consists of three parts: basic Information, decision rules, and reasoning rules.

**Basic Information.** This section covers general settings related to the underlying LLM, specifically including:

- **LLM Base Model Information**: Specifies the LLM API or the locally deployed LLM model to be used. The corresponding JSON object is as follows:

```
1  "llm_base_model": {
2    "model_type": "local",
```

```
3          "model_config_file_path": "/model/Llama-3-8B-
       Instruct/config.py"
4        }
```

Here, `model_type` constrains whether the model used is a local model or an API call. `local` represents the use of a local model, and `api` represents using an online LLM API. The `model_config_file_path` points to the LLM configuration file.

- **Maximum Token Count**: Sets the maximum number of tokens available for task execution. The corresponding JSON object is as follows:

```
1      "token_limits": {
2          "max_token_count": 4096
3        }
```

`max_token_count` refers to the maximum number of tokens allowed; exceeding this count will terminate the inference process.

- **Maximum Node Count**: Limits the maximum number of nodes allowed during the task execution process. The corresponding JSON object is as follows:

```
1      "structure_limits": {
2          "max_node_count": 500
3        }
```

Here, `max_node_count` refers to the maximum number of nodes allowed; exceeding this count will also terminate the inference process.

**Decision Rule.** This rule is used to determine how to decide the next operation given a specific node, specifically including:

- **Stop Judgment**: Defines the conditions under which the LLM's reasoning process should be terminated. The corresponding JSON object is as follows:

```
1      "stop_judgment": {
2          "condition": "Based on the information from the
       current node or the parent node, it is no longer
       possible to derive a calculation result of Game of 24,
       or the reasoning has entered a cycle."
3        }
```

- **Answer Judgment**: Specifies the conditions under which the content of the current node can be output as the final answer. The corresponding JSON object is as follows:

```
1      "answer_judgment": {
2          "condition": "The current method can calculate and
       obtain 24."
3        }
```

**Reasoning Rules.** Define the generation and connection rules to be followed if a node requires further expansion in reasoning. Each rule specifically includes:

- **Topological Judgment**: Includes the topological conditions that a node must satisfy for this rule to apply, such as the distance to the root node, node in-degree, the required subgraph structure, and a corresponding textual description. The corresponding JSON object is as follows:

```
1      "topological_judgment": {
2          "distance_from_root": [1,3],
```

```
3           "allowed_in_degree": [1,3],
4           "allowed_out_degree": [0,3],
5           "required_subgraph_structure": "It needs to have
     descendant nodes with a distance greater than 2."
6              }
```

`max_distance_from_root` constrains the range of the distance from the current node to the root node. $[1,3]$ represents the nodes that satisfy the current rule, where the distance to the root node (i.e., the shortest path to the root node) is between 1 and 3. If no such constraint is applied, the value is $[-1,-1]$. `allowed_in_degree` represents the allowed in-degree of the current node. Similarly, if there is no restriction, the value is $[-1,-1]$. `allowed_out_degree` follows the same logic. `required_subgraph_structure` represents the textual description of other possible graph structures required, which will be evaluated using the LLM for graph structure determination. If there is no restriction, the value is `N/A`.

- **Semantic Judgment**: Describes in textual form the semantic features that an applicable node should possess. The corresponding JSON object is as follows:

```
1           "semantic_judgment": {
2             "description": "Contain information about AI
     technology."
3              }
```

The `description` provides the semantic features that are used to determine whether this rule should be applied.

- $G_{sub}$ **Structure**: Specifies the composition of the generated set of child nodes $Ch(v)$ and its related set of parent nodes $Pa(Ch(v))$. If multiple nodes are generated using a single round of LLM interaction, the system will first generate one node. This node is then split based on predefined separation identifiers, and the associations between nodes are determined with the aid of the LLM to construct the edge structure. The corresponding JSON object is as follows:

```
1           "G_sub_structure": {
2             "child_nodes": {
3               "strategy": "single_round_multi_node",
4               "num_of_child_nodes": 1
5             },
6             "parent_nodes": {
7               "shortest_path_to_root": "include",
8               "sibling_node": "include",
9               "search_prompt": "N/A"
10            }
11          }
```

In `child_nodes`, the value of `strategy` can be either `single_round_multi_node` or `single_round_single_node`, which indicates whether a single round of LLM interaction generates all child nodes or just a single child node. `num_of_child_nodes` defines the number of child nodes to be generated. If `single_round_multi_node` is selected, this value remains 1, and the framework itself will handle the splitting of child nodes in subsequent steps.

In `parent_nodes`, `shortest_path_to_root` represents all the nodes in the shortest path from the current node to the root node. `sibling_node` indicates whether sibling nodes are included, with options `include` and `exclude`. If `search_prompt` is `N/A`, no search is conducted; otherwise, the parent nodes are searched according to the content of `search_prompt`.

- **Reasoning Prompt**: Provides the specific prompt text used.

Note that multiple reasoning rules will be defined, and for any given node requiring further reasoning, exactly one rule applies.

To provide a concrete example, we present the corresponding configuration file for the running example in Section 3.4. The content of this configuration file is shown below:

```
{
  "framework": "ToT for Game of 24",
  "configuration": {
    "basic_information": {
      "llm_base_model": {
        "model_type": "local",
        "model_config_file_path": "/model/Llama-3-8B-Instruct/
config.py"
      },
      "token_limits": {
        "max_token_count": 4096
      },
      "structure_limits": {
        "max_node_count": 500
      }
    },
    "decision_rules": {
      "stop_judgment": {
        "condition": "Based on the information from the current
node or the parent node, it is no longer possible to derive a
calculation result of Game of 24, or the reasoning has entered
 a cycle."
      },
      "answer_judgment": {
        "condition": "The current method can calculate and obtain
24."
      }
    },
    "reasoning_rules": [
      {
        "rule_id": "rule_for_root",
        "topological_judgment": {
          "max_distance_from_root": [0,0],
          "allowed_in_degree_range": [-1,-1],
          "required_subgraph_structure": "N/A"
        },
        "semantic_judgment": {
          "description": "N/A"
        },
        "G_sub_structure": {
          "child_nodes": {
            "strategy": "single_round_multi_node",
            "num_of_child_nodes": 1
          },
          "parent_nodes": {
            "shortest_path_to_root": "include",
            "sibling_node": "exclude",
            "search_prompt": "N/A"
          }
        },
        "reasoning_prompt": {
          "prompt_text": "You are a Game of 24 game expert. Please
 solve the given problem, provide 3 different reasoning nodes.
 Consider the most effective solving strategies. Note that
each approach should only advance one step, meaning only
compute one additional number. The current computed number
pool is: [ ]. Add two numbers to this pool. Output only the
combinations. Follow the format: option: [ ], updated number
pool: [ ]. Do not output any other content."
        }
      },
```

```
50        {
51          "rule_id": "rule_backtrack",
52          "topological_judgment": {
53            "max_distance_from_root": [-1,-1],
54            "allowed_in_degree_range": [-1,-1]],
55            "required_subgraph_structure": "N/A"
56          },
57          "semantic_judgment": {
58            "description": "N/A"
59          },
60          "G_sub_structure": {
61            "child_nodes": {
62              "strategy": "single_round_multi_node",
63              "num_of_child_nodes": 1
64            },
65            "parent_nodes": {
66              "shortest_path_to_root": "include",
67              "sibling_node": "exclude",
68              "search_prompt": "N/A"
69            }
70          },
71          "reasoning_prompt": {
72            "prompt_text": "You are a Game of 24 game expert. Please
      solve the given problem, provide 2 different reasoning nodes.
      Solve the problem from the parent node of the current node.
   Consider the most effective solving strategies. Note that each
      approach should only advance one step, meaning only compute
   one additional number. The current computed number pool is: [
   ]. Add two numbers to this pool. Output only the combinations.
      Follow the format: option: [ ], updated number pool: [ ]. Do
   not output any other content."
73          }
74        },
75        {
76          "rule_id": "rule_default",
77          "topological_judgment": {
78            "max_distance_from_root": [-1,-1],
79            "allowed_in_degree_range": [-1,-1],
80            "required_subgraph_structure": "N/A"
81          },
82          "semantic_judgment": {
83            "semantic_description": "N/A"
84          },
85          "G_sub_structure": {
86            "child_nodes": {
87              "strategy": "single_round_multi_node",
88              "num_of_child_nodes": 1
89            },
90            "parent_nodes": {
91              "shortest_path_to_root": "include",
92              "sibling_node": "exclude",
93              "search_prompt": "N/A"
94            }
95          },
96          "reasoning_prompt": {
97            "prompt_text": "You are a Game of 24 game expert. Please
      solve the given problem, provide 2 different reasoning nodes.
       Consider the most effective solving strategies. Note that
   each approach should only advance one step, meaning only
   compute one additional number. The current computed number
   pool is: [ ]. Add two numbers to this pool. Output only the
   combinations. Follow the format: option: [ ], updated number
   pool: [ ]. Do not output any other content."
98          }
99        }
```

```
100          ]
101        }
102  }
```

# D    PROOF OF PROPOSITION 1

**Proposition 1** *Kernel matrix $\boldsymbol{K}_{RGWL}^{(h)}$ of $\mathbb{K}_{RGWL}^{(h)}(\cdot,\cdot)$ is positive semi-definite (p.s.d.).*

*Proof.* According to the proposition, for $\boldsymbol{c} \in \mathbb{R}^n$, we have:

$$
\boldsymbol{c}^\top \boldsymbol{K}_{\text{RGWL}}^{(h)} \boldsymbol{c} = \sum_{i=1}^{M} \sum_{j=1}^{M} c_i c_j \mathbb{K}_{\text{RGWL}}^{(h)}(G_i, G_j)
$$

$$
= \sum_{i=1}^{M} \sum_{j=1}^{M} c_i c_j \Big( \sum_{i=1}^{h} \Big( \Big\langle \eta\left(\widetilde{\psi}^{(i)}\left(\rho\left(\tau(G_i)\right)\right)\right), \eta\left(\widetilde{\psi}^{(j)}\left(\rho\left(\tau(G_j)\right)\right)\right) \Big\rangle
$$

$$
+ \lambda_i \lambda_j \Big\langle \eta\left(\widetilde{\psi}^{(i)}\left(\rho\left(G_i\right)\right)\right), \eta\left(\widetilde{\psi}^{(j)}\left(\rho\left(G_j\right)\right)\right) \Big\rangle \Big) \Big)
$$

$$
= \sum_{i=1}^{M} \sum_{j=1}^{M} c_i c_j \Big( \sum_{i=1}^{h} \Big\langle \eta\left(\widetilde{\psi}^{(i)}\left(\rho\left(\tau(G_i)\right)\right)\right), \eta\left(\widetilde{\psi}^{(i)}\left(\rho\left(\tau(G_j)\right)\right)\right) \Big\rangle
$$

$$
+ \lambda_i \lambda_j \sum_{i=1}^{h} \Big\langle \eta\left(\widetilde{\psi}^{(i)}\left(\rho\left(G_i\right)\right)\right), \eta\left(\widetilde{\psi}^{(i)}\left(\rho\left(G_j\right)\right)\right) \Big\rangle \Big), \tag{6}
$$

where $\lambda_i$ denotes the very value of $\lambda$ according to $G_i$ and $G_j$. As $\rho(\cdot)$ only modifiy graph node features into labels, $\Big\langle \eta\left(\widetilde{\psi}^{(i)}\left(\rho\left(\tau(G_i)\right)\right)\right), \eta\left(\widetilde{\psi}^{(i)}\left(\rho\left(\tau(G_j)\right)\right)\right) \Big\rangle$ can be denoted as the inner product of certain vector $\alpha_i$ and $\alpha_j$, $\alpha_i \in \mathbb{N}_0^h$ and $\alpha_j \in \mathbb{N}_0^h$. Similarly, we denote $\Big\langle \eta\left(\widetilde{\psi}^{(i)}\left(\rho\left(G_i\right)\right)\right), \eta\left(\widetilde{\psi}^{(i)}\left(\rho\left(G_j\right)\right)\right) \Big\rangle$ as the inner product of $\beta_i$ and $\beta_j$, $\beta_i \in \mathbb{N}_0^h$ and $\beta_j \in \mathbb{N}_0^h$. Therefore, we have:

$$
\boldsymbol{c}^\top \boldsymbol{K}_{\text{RGWL}}^{(h)} \boldsymbol{c} = \sum_{i=1}^{M} \sum_{j=1}^{M} c_i c_j \Big( \langle \alpha_i, \alpha_j \rangle + \lambda_i \lambda_j \langle \beta_i, \beta_j \rangle \Big)
$$

$$
= \sum_{i=1}^{M} \sum_{j=1}^{M} \Big( c_i c_j \langle \alpha_i, \alpha_j \rangle \Big) + \sum_{i=1}^{M} \sum_{j=1}^{M} \Big( c_i c_j \lambda_i \lambda_j \langle \beta_i, \beta_j \rangle \Big)
$$

$$
= \Big\langle \sum_{i=1}^{M} c_i \alpha_i, \sum_{j=1}^{M} c_j \alpha_j \Big\rangle + \Big\langle \sum_{i=1}^{M} c_i \lambda_i \alpha_i, \sum_{j=1}^{M} c_j \lambda_j \alpha_j \Big\rangle
$$

$$
= \Big\| \sum_{i=1}^{M} c_i \alpha_i \Big\|^2 + \Big\| \sum_{i=1}^{M} c_i \lambda_i \alpha_i \Big\|^2 \geq 0. \tag{7}
$$

Based on the definition of positive semidefinite matrices, the proposition is proven.    $\square$

# E    DETIAIL IMPLEMENTATION OF $\tau(\cdot)$ AND $\rho(\cdot)$

The function $\tau(\cdot)$ can be simply identified and obtained from the paths in the graph. Specifically, we use a graph search algorithm to find all paths from the root to the answer node, and then we save all the nodes along these paths, along with the corresponding inference content.

$\rho(\cdot)$ characterizes the KNN-based graph node labeling. Initially, a large language model is employed to describe all relevant ideas in a consistent format. Subsequently, the textual responses associated with each node are embedded into feature vectors using a language model (Reimers & Gurevych,

2019). Following this, the Elbow Method (Cui et al., 2020) is applied to cluster the feature vector sets. The clustering process terminates once the Sum of Squared Errors (SSE) falls below a predefined threshold, denoted as $\delta$, which is treated as a hyperparameter. Nodes within the same cluster are then assigned the same label. Algorithm 2 demonstrates the procedure formally.

---

**Algorithm 2** KNN-based Graph Node Labeling with Elbow Method for Clustering

---

1: **Input:** Graph $G$ and $G'$ with nodes, large language model $LM$, predefined threshold $\delta$, feature embedding language model $f^{\text{LM}}(\cdot)$.
2: **Output:** Node labels
3: **1.** Use the large language model to describe all relevant ideas associated with each node $v$ within $G$ and $G'$ in a consistent format.
4: **2.** Embed the textual responses of each node $v$ into feature vectors using $f^{\text{LM}}(\cdot)$.
5: **3.** Apply the Elbow Method to cluster the feature vector sets:
6:     For each $k = 1, 2, \ldots,$ number of nodes:
7:         Compute the Sum of Squared Errors (SSE) for each clustering result.
8:         Terminate clustering when the SSE falls below the threshold $\delta$.
9: **5.** Assign the same label to all nodes within the same cluster.
10: **6.** Return the node labels.

---

## F    MORE REASONING FRAMEWORK IMPLEMENTATIONS AND WORKFLOWS

In this section, we provide more reasoning examples to better demonstrate TiG. We first present the configuration file and reasoning process used with the ToT framework on Legal Cases. The configuration file is as follows:

```
1  {
2    "framework": "ToT for Legal Case",
3    "configuration": {
4      "basic_information": {
5        "llm_base_model": {
6          "model_type": "local",
7          "model_config_file_path": "/model/gpt-4o/config.py"
8        },
9        "token_limits": {
10         "max_token_count": 4096
11       },
12       "structure_limits": {
13         "max_node_count": 500
14       }
15     },
16
17     "decision_rules": {
18       "stop_judgment": {
19         "condition": "Stop if the current node (or its parent) can
     no longer advance the theft vs. fraud distinction, the amount
      attribution is already fixed for the active branch, the path
     enters a loop (semantic or structural repetition), or the node
      repeats previously concluded legal reasoning without adding
     new statutory or factual analysis."
20       },
21       "answer_judgment": {
22         "condition": "The path reaches a final legal conclusion,
     the reasoning explicitly distinguishes the secret
     appropriation phase from the later deceitful transfer, and
     clarifies amount attribution for each offense."
23       }
24     },
25
26     "reasoning_rules": [
27
```

```
28        {
29          "rule_id": "rule_for_root",
30          "topological_judgment": {
31            "max_distance_from_root": [0, 0],
32            "allowed_in_degree_range": [0, 0],
33            "allowed_out_degree_range": [1, 3],
34            "required_subgraph_structure": "N/A"
35          },
36          "semantic_judgment": {
37            "description": "Applies only at the root question node
           of this case."
38          },
39          "G_sub_structure": {
40            "child_nodes": {
41              "strategy": "single_round_multi_node",
42              "num_of_child_nodes": 1
43            },
44            "parent_nodes": {
45              "shortest_path_to_root": "include",
46              "sibling_node": "exclude",
47              "search_prompt": "N/A"
48            }
49          },
50          "reasoning_prompt": {
51            "prompt_text": "You are a legal expert in Chinese
           Criminal Law. Instruction: do not assume fixed statute numbers
           . Instead, identify and retrieve the most relevant provisions
           of the Criminal Law by analyzing the conduct. Output strictly
           in the following format:\n reasoning: (one-sentence
           preliminary conclusion)."
52          }
53        },
54
55        {
56          "rule_id": "rule_theft_analysis",
57          "topological_judgment": {
58            "max_distance_from_root": [1, 2],
59            "allowed_in_degree_range": [1, 3],
60            "allowed_out_degree_range": [0, 3],
61            "required_subgraph_structure": "The path to root
           includes a node that flagged D1 (theft analysis) as an active
           direction."
62          },
63          "semantic_judgment": {
64            "description": "Node focuses on secret taking via
           impersonation; evaluates whether conduct fits theft-related
           statutory elements (secret appropriation of another's property
           )."
65          },
66          "G_sub_structure": {
67            "child_nodes": {
68              "strategy": "single_round_multi_node",
69              "num_of_child_nodes": 1
70            },
71            "parent_nodes": {
72              "shortest_path_to_root": "include",
73              "sibling_node": "exclude",
74              "search_prompt": "exclude"
75            }
76          },
77          "reasoning_prompt": {
78            "prompt_text": "Advance ONE step on theft analysis for
           the secret impersonation/loan-obtaining phase. Identify
           relevant statutory elements without assuming specific article
           numbers. Map facts to elements: (i) secrecy, (ii)
```

```
1458            appropriation, (iii) object = other's property, (iv) intent to
1459             unlawfully possess. Then produce sub-conclusions depending on
1460             money ownership at the moment of appropriation (bank vs
1461            victim). Output strictly: \n reasoning: (element-wise mapping
1462            in one sentence).\n (No extra text.)"
     79            }
1463   80          },
1464   81
1465   82          {
1466   83            "rule_id": "rule_fraud_analysis",
1467   84            "topological_judgment": {
1468   85              "max_distance_from_root": [1, 2],
1469   86              "allowed_in_degree_range": [1, 3],
1470   87              "allowed_out_degree_range": [0, 3],
       88              "required_subgraph_structure": "The path to root
1471         includes a node that flagged D2 (fraud analysis) as an active
1472         direction."
1473   89            },
       90            "semantic_judgment": {
1474   91              "description": "Node focuses on deceit-induced transfer;
1475          evaluates whether conduct fits fraud-related statutory
1476         elements (obtaining property by deception)."
1477   92            },
1478   93            "G_sub_structure": {
1479   94              "child_nodes": {
       95                "strategy": "single_round_multi_node",
1480   96                "num_of_child_nodes": 1
1481   97              },
1482   98              "parent_nodes": {
       99                "shortest_path_to_root": "include",
1483   100               "sibling_node": "exclude",
1484   101               "search_prompt": "exclude"
1485   102             }
1486   103           },
1487   104           "reasoning_prompt": {
       105             "prompt_text": "Advance ONE step on fraud analysis for
1488         the later inducement/transfer phase. Identify relevant
1489         statutory elements without assuming specific article numbers.
1490         Map facts to elements: (i) false representation/concealment, (
1491         ii) victim's disposal of property, (iii) causal link, (iv)
1492         unlawful possession. Then produce sub-conclusions depending on
1493          the victim's disposal awareness (fully misled vs partially
1494         aware). Output strictly:\n reasoning: (element-wise mapping in
1495          one sentence).\n (No extra text.)"
       106             }
1496   107         },
1497   108
1498   109         {
       110           "rule_id": "rule_amount_attribution_and_concurrence",
1499   111           "topological_judgment": {
1500   112             "max_distance_from_root": [2, 4],
1501   113             "allowed_in_degree_range": [1, 5],
1502   114             "allowed_out_degree_range": [0, 3],
       115             "required_subgraph_structure": "The ancestor chain must
1503         already contain at least one theft-focused node and one fraud-
1504         focused node."
1505   116           },
1506   117           "semantic_judgment": {
       118             "description": "Node determines the ownership/
1507         attribution of the loan at each timepoint and whether theft
1508         and fraud should be punished cumulatively."
1509   119           },
1510   120           "G_sub_structure": {
1511   121             "child_nodes": {
       122               "strategy": "single_round_single_node",
```

```
123           "num_of_child_nodes": 1
124         },
125         "parent_nodes": {
126           "shortest_path_to_root": "include",
127           "sibling_node": "include",
128           "search_prompt": "Summon the two closest analysis
      nodes (theft and fraud) on the shortest path for joint
      synthesis."
129         }
130       },
131       "reasoning_prompt": {
132         "prompt_text": "Synthesize the results of theft and
      fraud analyses. Decide (i) at impersonation moment the loaned
      money is owned by bank or victim, (ii) the later induced
      transfer disposes of <victim>'s property by deception, and (
      iii) whether the offenses concur and shall be combined for
      punishment. Output strictly:\n reasoning: (one-sentence
      rationale linking timepoints to ownership) \n  (No extra text
      .)"
133       }
134     },
135
136     {
137       "rule_id": "rule_finalize",
138       "topological_judgment": {
139         "max_distance_from_root": [2, 6],
140         "allowed_in_degree_range": [1, 10],
141         "allowed_out_degree_range": [0, 1],
142         "required_subgraph_structure": "Upstream nodes already
      fixed both: (i) theft elements satisfied for impersonation
      stage; (ii) fraud elements satisfied for inducement stage; and
       amount attribution coherent."
143       },
144       "semantic_judgment": {
145         "description": "All material elements established."
146       },
147       "G_sub_structure": {
148         "child_nodes": {
149           "strategy": "single_round_single_node",
150           "num_of_child_nodes": 1
151         },
152         "parent_nodes": {
153           "shortest_path_to_root": "include",
154           "sibling_node": "exclude",
155           "search_prompt": "N/A"
156         }
157       },
158       "reasoning_prompt": {
159         "prompt_text": "Issue the final conclusion exactly as
      the higher court maintained: theft + fraud, combined
      punishment. Also state in one sentence the amount attribution
      logic (impersonation stage vs later induced transfer). Output
      strictly:\n final: (theft + fraud; sentences combined True)\n
      rationale: (one-sentence amount-attribution and concurrence
      explanation)\n (No extra text.)"
160       }
161     }
162
163   ]
164  }
165 }
```

Next, we present the specific outputs of the model, including textual descriptions of the reasoning graphs and the content of each reasoning node.

---

**Example Output**

**Graph structure:**

Nodes:
[t=0] :root 0
[t=1] :node 1 node 2 node 3
[t=2] :node 4 node 5 node 6 node 7 node 8 node 9
[t=3] :node 10 node 11 node 12 node 13

Edges:
root 0 → node 1
root 0 → node 2
root 0 → node 3
node 1 → node 4
node 1 → node 5
node 2 → node 6
node 2 → node 7
node 3 → node 8
node 3 → node 9
node 5 → node 10
node 5 → node 11
node 7 → node 12
node 9 → node 13

Terminated nodes:
node 4, node 6, node 8, node 10, node 11, node 12

Answer node:
node 13

Answer path:
root 0 → node 3 → node 9 → node 13

**Reasoning Procedure:**

[t=1] Nodes:

node 1:
reasoning: Secret impersonation of the victim at the loan stage may constitute theft-related statutory elements. next_check: Ownership of loan funds (bank or victim).
Rule applied: rule_for_root.
Why proceed: This branch explores theft-related analysis.

node 2:
reasoning: Inducing the victim to transfer funds later may constitute fraud-related statutory elements.
next_check: Victim's awareness and property disposal.
Rule applied: rule_for_root.
Why proceed: This branch explores fraud-related analysis.

node 3:
reasoning: Determination of ownership of loan funds is crucial for deciding theft vs. fraud and concurrence.

---

next_check: Attribution of property and concurrence rules. Rule applied: rule_for_root.
Why proceed: This branch explores amount attribution and concurrence.

[t=2] Nodes:

node 4 (terminated False):
reasoning: If loan funds are considered victim-owned, appropriation = theft True element mapping.
branch: victim-owned
next_check: N/A
Rule applied: rule_theft_analysis.
Termination reason: No further statutory issues to check once victim-ownership path is exhausted. Marked False (terminated).

node 5:
reasoning: If loan funds are considered bank-owned, appropriation analysis continues.
branch: bank-owned
next_check: Further confirmation of ownership and appropriation.
Rule applied: rule_theft_analysis.
Why proceed: Ownership requires deeper analysis, not terminated.

node 6 (terminated False):
reasoning: If the victim is fully deceived and disposes of property, fraud-related elements are satisfied. branch: fully-misled next_check: N/A Rule applied: rule_fraud_analysis. Termination reason: Complete satisfaction of fraud elements. Marked False.

node 7:
reasoning: If the victim is partially aware but still induced to transfer funds, fraud requires concurrence analysis.
branch: partially-aware
next_check: Offense concurrence and attribution.
Rule applied: rule_fraud_analysis.
Why proceed: Requires deeper concurrence analysis.

node 8 (terminated False):
reasoning: If loan funds entirely belong to the victim, only theft is applicable.
conclusion: single offense False
next_check: N/A
Rule applied: rule_amount_attribution_and_concurrence.
Termination reason: This conclusion contradicts the higher court (single offense). Terminated False.

node 9:
reasoning: If loan funds initially belong to the bank, later deceitful transfer is fraud, leading to combined punishment.
conclusion: pending final check
next_check: Final concurrence confirmation.
Rule applied: rule_amount_attribution_and_concurrence (from config).
Why proceed: This aligns with the higher court's reasoning, must continue.

[t=3] Nodes:

node 10 (terminated False):
reasoning: Bank retains ownership; impersonation = theft.
conclusion: theft established True
next_check: N/A
Rule applied: rule_theft_analysis.

Termination reason: Stopped at theft-only branch, contradicts combined judgment → False.

node 11 (terminated False):
reasoning: Loan as debt only, not property; theft not established.
conclusion: theft not False
next_check: N/A
Rule applied: rule_theft_analysis.
Termination reason: Theft elements negated → branch ends False.

node 12 (terminated False):
reasoning: Victim partially aware, inducement insufficient → fraud weakened.
conclusion: fraud not False
next_check: N/A
Rule applied: rule_fraud_analysis (from config).
Termination reason: Fraud elements not fully satisfied → branch ends False.

node 13 (True final answer):
final: theft + fraud; sentences combined True
rationale: At impersonation stage, appropriation = theft against bank's property; at inducement stage, transfer = fraud against victim's property; ownership attribution supports two offenses in concurrence.
Rule applied: rule_finalize (from config).
Why valid: Matches higher court judgment; distinguishes phases and attributes amounts correctly → True

Next, we present an example using the GoT framework on College Entrance Examination Questions. The configuration file is as follows:

```json
{
  "framework": "GoT for College Entrance Examination Question",
  "configuration": {
    "basic_information": {
      "llm_base_model": {
        "model_type": "local",
        "model_config_file_path": "/model/gpt-4o/config.py"
      },
      "token_limits": {
        "max_token_count": 4096
      },
      "structure_limits": {
        "max_node_count": 500
      }
    },

    "decision_rules": {
      "stop_judgment": {
        "condition": "Stop if the current node (or its parent)
    cannot further advance the mathematical reasoning, the branch
    yields contradictions, or all solution cases are exhausted."
      },
      "answer_judgment": {
        "condition": "The path reaches a final conclusion
    consistent with the expected solution form (numeric range,
    interval, or multiple-choice)."
      }
    },

    "reasoning_rules": [

      {
        "rule_id": "rule_for_root",
```

```
30          "topological_judgment": {
31            "max_distance_from_root": [0, 0],
32            "allowed_in_degree_range": [0, 0],
33            "allowed_out_degree_range": [1, 3],
34            "required_subgraph_structure": "N/A"
35          },
36          "semantic_judgment": {
37            "description": "Root node: classifies the math problem
      and generates initial reasoning directions."
38          },
39          "G_sub_structure": {
40            "child_nodes": {
41              "strategy": "single_round_multi_node",
42              "num_of_child_nodes": 3
43            },
44            "parent_nodes": {
45              "shortest_path_to_root": "include",
46              "sibling_node": "exclude",
47              "search_prompt": "N/A"
48            }
49          },
50          "reasoning_prompt": {
51              "prompt_text": "You are a math expert. Generate high-
      level reasoning direction advancing ONE step: Output strictly:
       \n reasoning: (one-sentence preliminary step)."
52
53          }
54        },
55
56        {
57          "rule_id": "rule_equation_analysis",
58          "topological_judgment": {
59            "max_distance_from_root": [1, 3],
60            "allowed_in_degree_range": [1, 3],
61            "allowed_out_degree_range": [0, 3],
62            "required_subgraph_structure": "The path to root
      includes a node flagged D1."
63          },
64          "semantic_judgment": {
65            "description": "Analyzes quadratic equations (
      discriminant, factorization, or root conditions)."
66          },
67          "G_sub_structure": {
68            "child_nodes": {
69              "strategy": "single_round_multi_node",
70              "num_of_child_nodes": 2
71            },
72            "parent_nodes": {
73              "shortest_path_to_root": "include",
74              "sibling_node": "exclude",
75              "search_prompt": "Retrieve ancestor node introducing
      equation analysis."
76            }
77          },
78          "reasoning_prompt": {
79              "prompt_text": "Advance ONE step in equation/roots
      analysis. Output strictly:\n reasoning: (root/condition
      analysis in one sentence)."
80          }
81        },
82
83        {
84          "rule_id": "rule_set_operations",
85          "topological_judgment": {
86            "max_distance_from_root": [1, 3],
```

```
 87              "allowed_in_degree_range": [1, 3],
 88              "allowed_out_degree_range": [0, 3],
 89              "required_subgraph_structure": "The path to root
        includes a node flagged D2."
 90            },
 91            "semantic_judgment": {
 92              "description": "Analyzes subset and inclusion relations
        between sets."
 93            },
 94            "G_sub_structure": {
 95              "child_nodes": {
 96                "strategy": "single_round_multi_node",
 97                "num_of_child_nodes": 2
 98              },
 99              "parent_nodes": {
100                "shortest_path_to_root": "include",
101                "sibling_node": "exclude",
102                "search_prompt": "Retrieve ancestor node introducing
        set operation analysis."
103              }
104            },
105            "reasoning_prompt": {
106              "prompt_text": "Advance ONE step in analysis. Output
        strictly:\n reasoning: (analysis). "
107            }
108          },
109
110          {
111            "rule_id": "rule_backtrack",
112            "topological_judgment": {
113              "max_distance_from_root": [-1, -1],
114              "allowed_in_degree_range": [1, 10],
115              "allowed_out_degree_range": [0, 2],
116              "required_subgraph_structure": "N/A"
117            },
118            "semantic_judgment": {
119              "description": "The branch stalls or contradicts."
120            },
121            "G_sub_structure": {
122              "child_nodes": {
123                "strategy": "single_round_single_node",
124                "num_of_child_nodes": 1
125              },
126              "parent_nodes": {
127                "shortest_path_to_root": "include",
128                "sibling_node": "exclude",
129                "search_prompt": "Find the closest ancestor
        introducing the conflicting assumption."
130              }
131            },
132            "reasoning_prompt": {
133              "prompt_text": "Backtrack ONE step to the nearest
        ancestor with a wrong or incomplete assumption. Propose a
        minimally revised condition. Output strictly:\n reasoning: (
        short fix)."
134            }
135          },
136
137          {
138            "rule_id": "rule_finalize",
139            "topological_judgment": {
140              "max_distance_from_root": [2, 6],
141              "allowed_in_degree_range": [1, 10],
142              "allowed_out_degree_range": [0, 1],
```

```
143            "required_subgraph_structure": "Upstream nodes already
         fixed both: (i) equation root conditions, (ii) set-inclusion
         conditions."
144            },
145            "semantic_judgment": {
146              "description": "All relevant conditions are satisfied;
         issue the final solution."
147            },
148            "G_sub_structure": {
149              "child_nodes": {
150                "strategy": "single_round_single_node",
151                "num_of_child_nodes": 1
152              },
153              "parent_nodes": {
154                "shortest_path_to_root": "include",
155                "sibling_node": "exclude",
156                "search_prompt": "N/A"
157              }
158            },
159            "reasoning_prompt": {
160              "prompt_text": "Issue the final answer strictly in the
         problem's required format. Output strictly:\n final: (answer)
         \n rationale: (one-sentence justification) \n (No extra text.)
         "
161            }
162          }
163
164        ]
165      }
166 }
```

Next, we present the specific outputs of the model, including textual descriptions of the reasoning graphs and the content of each reasoning node.

---

**Example Output**

**Graph structure:**

Nodes:
[t=0] :root 0
[t=1] :node 1 node 2 node 3
[t=2] :node 4 node 5 node 6 node 7
[t=3] :node 8 node 9 node 10

Edges:
root 0 → node 1
root 0 → node 2
root 0 → node 3
node 2 → node 4
node 2 → node 5
node 3 → node 8
node 8 → node 3 (backtracking)
node 3 → node 9
node 4 → node 8
node 5 → node 9
node 7 → node 10

Terminated nodes:
node 1, node 8, node 9

---

Answer node:
node 10

Answer path:
root 0 → node 3 → node 7 → node 10

**Reasoning Procedure:**

node 1 (terminated False):
reasoning: Solve roots of $A$ and $B$; check discriminant of $B$.
next_check: Whether roots of $B$ lie inside $\{1, 2\}$.
Rule applied: rule_for_root.
Termination reason: Incomplete, does not yet impose subset condition → False.

node 2:
reasoning: For the condition $A \cup B = A$, require $B \subseteq A$.
next_check: Compare roots of $B$ with elements of $A$.
Rule applied: rule_for_root.
Why proceed: Core set-inclusion condition.

node 3:
reasoning: Consistency check: verify parameter $a$ effect on roots of $B$.
next_check: Analyze $a$ values to ensure $B \subseteq A$.
Rule applied: rule_for_root.
Why proceed: Directly links to final inclusion condition.

node 4:
reasoning: Roots of $B$: $x = 1$ and $x = a - 1$.
branch: case1: both roots in $A$ — case2: one root outside $A$
next_check: Whether $a - 1 \in \{1, 2\}$.
Rule applied: rule_equation_analysis.

node 8 (terminated False):
reasoning: If $a - 1 \notin \{1, 2\}$, then $B$ contains elements not in $A$.
conclusion: False
Rule applied: rule_equation_analysis.
Termination reason: Contradicts requirement $B \subseteq A$ → False.

node 5:
reasoning: Require $a - 1$ equals 1 or 2, so $a = 2$ or $a = 3$.
branch: case1: $a = 2$ — case2: $a = 3$
next_check: Check consistency with discriminant.
Rule applied: rule_set_operations.

node 9 (terminated False):
reasoning: If $a = 3$, discriminant fails, contradiction.
conclusion: False
Rule applied: rule_set_operations.
Termination reason: Inconsistent with quadratic constraints → False.

node 6 (backtrack):
reasoning: Reconsider condition $a - 1 \in \{1, 2\}$. Correct range: $1 \leq a \leq 2$.
backtrack_to: node 3
next_check: Re-evaluate subset condition with corrected parameter range.
Rule applied: rule_backtrack.

node 7:

reasoning: Within $1 \leq a \leq 2$, roots of $B$ are in $\{1, 2\}$, so $B \subseteq A$.
conclusion: supports condition
next_check: Finalize answer.
Rule applied: rule_set_operations.

node 10 (True final answer):
final: $1 \leq a \leq 2$
rationale: Within $1 \leq a \leq 2$, $B \subseteq A$ holds, hence $A \cup B = A$.
Rule applied: rule_finalize.
Why valid: Matches expected answer $\rightarrow$ True

