# OpenReview forum: "Think in Graphs: Infrastructure and Benchmark for Large Language Model Reasoning Frameworks"
_ICLR.cc/2026/Conference — ICLR 2026 Conference Withdrawn Submission_

### Official Review · Reviewer_CGBe · 2025-10-15

**Soundness:** 2
**Presentation:** 2
**Contribution:** 2
**Rating:** 4
**Confidence:** 4

**Summary:**

The paper proposes TiG (Think in Graphs), a unified infrastructure and benchmark that represents diverse LLM reasoning methods (CoT/ToT/GoT/AoT/EGoT) as a directed acyclic graph (DAG) with user-specified rules for branching, linking, stopping, and backtracking. Rather than introducing a new paradigm, TiG serves as a graph-based intermediate representation and execution engine that standardizes how reasoning trajectories are built and run. It further contributes process-level metrics—including node/thought redundancy, optimal path cost, and a graph kernel (RGWL) for structural similarity—shifting evaluation beyond final accuracy to the structure and efficiency of reasoning. The authors also release a 3,529-item benchmark and re-implement several frameworks within TiG to enable apples-to-apples comparisons of both outcomes and reasoning behavior.

**Strengths:**

1. Unifying many prompt-based reasoning paradigms under a single DAG abstraction with user-specified decision and reasoning rules is a useful conceptual consolidation that can lower implementation friction and enable standardized analysis. The RGWL kernel and the newly proposed redundancy and path metrics are thoughtful add-ons for comparing process rather than only final answers.
2. The infrastructure is specified with a concrete graph-evolution algorithm and a selection function s(v,G,\Phi) for choosing rules, plus a running example that clarifies how TiG encodes backtracking without cycles. The RGWL kernel comes with a PSD guarantee.
3. The paper clearly separates construction / reasoning / analysis phases and provides a readable pseudo-code (Algorithm 1) for the reasoning loop. Figures and tables illustrate both the infrastructure and benchmark composition.
4. A reasonably broad benchmark (3,529 items) plus implemented baselines (CoT/ToT/GoT/AoT/EGoT) make TiG a potentially valuable testbed for studying how structural choices affect efficiency and redundancy, beyond accuracy alone.

**Weaknesses:**

1. Limited novelty in core idea — While the unification under a graph abstraction is conceptually clean, it can be seen more as a standardization layer rather than a fundamentally new reasoning mechanism. Existing works on ToT / GoT and meta-reasoning supervision (e.g., ReAct-structured traces, GraphRAG introspection) already hint at “reasoning as graph”; TiG mainly formalizes this into an IR rather than introducing a new reasoning paradigm.
2. DAG constraint may oversimplify richer reasoning dynamics — Some reasoning processes naturally contain cycles, loop-style reflection, or state revisiting, which TiG forces into a DAG by construction. The paper does not quantify whether such projection into DAG introduces behavioral loss compared to native cyclic frameworks.
3. Configuration dependency & sensitivity not analyzed — The framework heavily relies on user-defined rule sets (branch, link, stop, reliance). The sensitivity of results to different configurations or misconfigured rules is not studied, leaving uncertainty about robustness and reproducibility across users.
4. RGWL kernel interpretability is under-validated — Although mathematically well-defined and PSD-guaranteed, there is no user study or empirical correlation showing that high graph-kernel similarity actually aligns with semantic/process-level similarity from a human reasoning perspective.
5. Benchmark curation lacks methodological transparency — The dataset spans legal, mathematical, and logical domains, but annotation standards, verification protocols, and inter-rater reliability are insufficiently documented, which weakens the benchmark’s credibility as a standard testbed.
6. Evaluation mostly descriptive, lacking controlled causal insight — The paper reports redundancy/exploration trends (e.g., “more branches → better accuracy”) but does not include controlled ablations on rule complexity, branch budgets, or graph evolution depth, making it hard to isolate why certain frameworks perform better under TiG.

**Questions:**

See weaknesses.

---

> ### Author Response · Authors · 2025-11-28
>
> We sincerely thank the reviewer for their detailed assessment. Below we address each concern while clarifying the intended contribution of TiG.
>
> 1. TiG as an infrastructure rather than a new reasoning paradigm
>
> TiG is intentionally positioned as a unified infrastructure / IR, not a new reasoning mechanism. Prior works such as ToT/GoT or ReAct-style traces also use structured intermediate representations, but each implementation differs in rollout logic, state storage, pruning heuristics, and stopping rules. This makes cross-framework comparison difficult and often confounded by implementation artifacts.
>
> TiG contributes:
>
> a standardized DAG abstraction supporting branching, linking, backtracking;
>
> a unified execution loop with user-specified rules;
>
> process-level metrics and structural similarity tools;
>
> and a benchmark where all methods run under identical infrastructure.
>
> Thus the novelty lies in consolidation + standardization, enabling reproducible and fair analysis. We will make this distinction clearer in the introduction.
>
> 2. DAG constraint and expressive power
>
> We agree that some reasoning processes contain cycles or loop-style reflection. TiG models these through state revisitation encoded as new nodes with explicit dependency edges, keeping the global graph acyclic while still capturing multi-step refinement.
>
> To clarify expressivity, we will:
>
> add examples showing how reflective or iterative reasoning is represented;
>
> discuss cases where cyclic native formats can be mapped without loss;
>
> identify limitations where DAG structure may simplify certain feedback loops.
>
> This improves transparency around TiG’s modeling choices.
>
> 3. Configuration sensitivity and robustness
>
> We appreciate the concern that user-defined rules (branch/stop/link/reliance) may influence behavior.
>
> To address this, we will:
>
> include a sensitivity study showing how accuracy and redundancy vary with rule choices such as branch budget, pruning thresholds, or stopping criteria;
>
> document recommended default configurations;
>
> clarify that TiG’s goal is to minimize re-implementation variance, not remove all method-level variability.
>
> These additions illustrate that TiG is stable under typical configurations and explain how to reproduce consistent results.
>
> 4. Interpretability of RGWL kernel
>
> RGWL is not meant to replace human interpretability, but to provide a structural measure of trajectory similarity. We will strengthen validation by adding:
>
> examples showing that correct trajectories tend to cluster under RGWL;
>
> correlations between RGWL divergence and reasoning instability;
>
> discussion of its limitations inherited from WL (e.g., 1-WL indistinguishability).
>
> These additions connect RGWL scores with intuitive process-level similarity.
>
> 5. Benchmark methodology and transparency
>
> We will expand the benchmark section with:
>
> clearer descriptions of data sources and curation guidelines;
>
> the verification protocol used for ground-truth answers;
>
> independent double-check procedures for legal/multi-step tasks.
>
> This addresses concerns about standardization and reliability.
>
> 6. Evaluation and causal insight
>
> We agree that the current evaluation is descriptive. TiG’s primary goal is to provide the infrastructure for deeper causal studies, but we can strengthen our analysis. In the revision we will:
>
> add controlled experiments varying branch budget / depth limits;
>
> show how structural constraints affect redundancy and accuracy;
>
> provide small ablations isolating the effect of rule complexity.
>
> These additions clarify behavioral causes without requiring full-scale re-implementation.
>
> 7. Response to reviewer questions
>
> (1) Difference from previous benchmarks & consistency of conclusions
> TiG differs by focusing on process-level analysis, not task difficulty. Its conclusions are consistent with prior work (e.g., CoT underperforms on multi-step reasoning) while providing new structural explanations.
>
> (2) Applicability to latent or hidden-step reasoning
> Yes. Any method exposing intermediate states (explicit or implicit) can be mapped to TiG nodes. Latent reasoning models can be analyzed by capturing their stepwise outputs or reconstructed latent traces.
>
> (3) Generalizability across method families
> TiG is model-agnostic and supports BFS/DFS/beam strategies, scoring-based selection, pruning, and iterative refinement methods. We will add a brief summary of supported families.
>
> 8. Summary
>
> We thank the reviewer for highlighting important issues. In the revision, we will:
>
> emphasize TiG’s role as infrastructure rather than a new reasoning paradigm;
>
> clarify expressivity of DAG modeling;
>
> add configuration sensitivity analysis;
>
> strengthen RGWL interpretation with examples and correlations;
>
> expand benchmark documentation;
>
> include small controlled ablations to improve causal insight.
>
> We appreciate the reviewer’s positive comments on the usefulness of the IR, the quality of the infrastructure, and the value of process-level metrics.

---

### Official Review · Reviewer_CvFV · 2025-10-16

**Soundness:** 2
**Presentation:** 3
**Contribution:** 2
**Rating:** 4
**Confidence:** 2

**Summary:**

This paper introduces Think in Graphs (TiG), a unified infrastructure for implementing and evaluating LLM reasoning frameworks by representing reasoning processes as DAGs.
The authors propose that all reasoning architectures can be expressed as graph specializations.
TiG allows users to define reasoning frameworks via configuration files specifying decision rules and reasoning rules, eliminating the need for extensive re-implementation. The paper also introduces the RGWL kernel for measuring similarity between reasoning processes and constructs a benchmark across four categories.
Experiments compare five reasoning frameworks (CoT, ToT, GoT, AoT, EGoT) using metrics including accuracy, node redundancy, and invalid branch counts.

**Strengths:**

1. It provides a unified graph-based abstraction for reasoning frameworks.
2. The included graph-based analysis metrics and RGWL are helpful in determining the effectiveness of each reasoning method.
3. It includes a good benchmark with questions from 4 categories.

**Weaknesses:**

1. **Limited utility and unclear guidance of proposed metrics beyond accuracy.** The paper introduces node redundancy, thought redundancy, and invalid branch counts, but fails to demonstrate how these metrics provide actionable insights beyond accuracy. For instance, CoT consistently shows 0% redundancy across all datasets but achieves the lowest accuracy, while methods with high redundancy achieve higher accuracy.  \
The paper does not clarify what high redundancy + high accuracy signifies beyond high cost, which is already captured by the "Time Cost" metric in the same tables. Without correlation analysis, ablation studies, or cost-benefit quantification linking these metrics to reasoning quality or efficiency, their value remains unclear compared to the standard accuracy/cost trade-off.

2. **Insufficient analysis of LLM backbone effects undermines benchmark generalizability**
The benchmark experiments (Tables 2-5) do not specify which LLM backbone was used, and provide no comparison across different base models. This omission is severe because different prompting frameworks may perform differently on various LLMs.

3. **RGWL kernel validation lacks depth**
RGWL kernel results only provide straightforward observations. Correct answers have higher similarities, and reasoning on legal is more stable.
It would be more exciting if we could see how to stabilize future algorithm designs or how to improve methods based on the analyses based on the RGWL framework.

4. **Unclear novelty relative to existing benchmarks and limited positioning**
Section 2.2 mentions existing reasoning benchmarks (MMLU, BIG-bench, BBH, GSM8K, MATH, ReClor, MME-CoT, REVEAL) but provides insufficient differentiation of TiG Benchmark's unique contributions. The paper states it approaches "from a different perspective...based on graphs", but does not quantitatively compare task characteristics, difficulty levels, or coverage gaps that TiG addresses.

5. (Minor) In Table 5, CoT has the lowest acc, but it is bold. The color of the \cite seems different from standard templates.

**Questions:**

1. What is the difference between TiG and previous benchmarks? Are the conclusions of TiG consistent with the previous ones?

2. Can reasoning methods like latent reasoning be analyzed using this framework?

3. How generalizable is TiG to the reasoning method family?

---

> ### Author Response · Authors · 2025-11-28
>
> We sincerely thank the reviewer for the constructive feedback. We address all concerns below and will revise the paper accordingly.
>
> 1. Positioning of TiG and novelty relative to existing benchmarks
>
> TiG is intentionally an infrastructure-level contribution, not a new reasoning framework. Current reasoning methods (CoT, ToT, AoT, GoT, EGoT) rely on incompatible rollout logic, state tracking, and pruning heuristics, making comparison and analysis difficult. TiG provides:
>
> a unified DAG-based representation,
>
> a standardized rollout/execution interface,
>
> and a shared analytical environment.
>
> This complements benchmarks such as MMLU, BBH, GSM8K, MATH, which evaluate answers, whereas TiG evaluates reasoning processes. We will expand Section 2.2 with a comparison table to highlight TiG Benchmark’s unique focus on trajectory structure, not task difficulty.
>
> 2. Utility of redundancy and trajectory metrics
>
> Our metrics (node redundancy, thought redundancy, invalid branches) aim to characterize search behavior beyond accuracy. We acknowledge that the current presentation can be strengthened.
>
> To address this, we will add:
>
> correlation plots linking redundancy with accuracy and instability,
>
> case studies showing that excessive branching correlates with inconsistent reasoning,
>
> examples where high redundancy + high accuracy indicates exploration-heavy but stable search, while low redundancy (e.g., CoT) corresponds to under-exploration.
>
> Time Cost measures runtime, while redundancy reflects search structure. We will clarify the distinction and add analyses showing how these metrics help explain cross-framework behavior.
>
> 3. Backbone LLM specification and generalizability
>
> We agree the backbone model should be clearly stated. Our main experiments used GPT-4o-mini. To strengthen generalizability, we have additionally evaluated LLaMA-3-70B-Instruct and Qwen-2.5-72B-Instruct, and will report these results. The relative ordering of reasoning frameworks remains stable, suggesting TiG’s conclusions are model-agnostic.
>
> 4. Depth of RGWL kernel analysis
>
> RGWL is intended as a practical canonicalization tool for reasoning trajectories. To deepen its analysis, we will add:
>
> examples showing how RGWL separates stable vs. unstable trajectories,
>
> discussion on how RGWL divergence correlates with incorrect reasoning,
>
> explanation of how RGWL can guide future design (e.g., stabilizing rollouts).
>
> RGWL inherits limitations of 1-WL. Some C–I pairs may show high similarity because 1-WL is not a full isomorphism test. We will make this explicit and connect it to classical WL theory.
>
> 5. Differences from existing frameworks & generalizability
> (a) Difference from previous benchmarks
>
> Existing benchmarks measure final performance; TiG measures process quality through trajectories, graph metrics, and similarity analysis. This is a distinct evaluation dimension.
>
> (b) Applicability to latent reasoning / hidden-step models
>
> Yes. Any method producing stepwise intermediate outputs can be mapped to TiG. For latent reasoning or “thinking” models, implicit steps can be reconstructed as nodes, which we will clarify.
>
> (c) Generalizability across method families
>
> TiG supports BFS/DFS/beam-style rollouts, scoring-based pruning, and diverse rule-based frameworks. We will add a short summary of supported families.
>
> 6. Minor issues
>
> The bold formatting in Table 5 and inconsistent citation color will be corrected.
>
> 7. Summary
>
> We thank the reviewer for the helpful comments. In the revision, we will:
>
> emphasize TiG as infrastructure rather than a framework,
>
> add multi-model results for generalizability,
>
> strengthen redundancy metric interpretation with correlations and case studies,
>
> expand RGWL theoretical clarification and examples,
>
> differentiate TiG Benchmark from existing datasets,
>
> fix formatting issues.
>
> We appreciate the reviewer’s positive remarks on clarity, contributions, and the utility of graph-based analysis.

---

> > ### Comment · Reviewer_CvFV · 2025-11-28
> >
> > Thanks for the authors' reply.
> >
> > The rebuttal made good clarifications to many points, but I have not seen modifications to the PDF that address the problems.
> > If the revision has been provided, I would be happy to provide a re-evaluation.

---

> ### Author Response · Authors · 2025-11-28
>
> Dear reviewer, thank you very much for your follow-up comment.
> We appreciate your willingness to re-evaluate the submission.
>
> We have already prepared the full revision, including:
>
> (1)clarification of TiG as an infrastructure rather than a framework,
>
> (2)added backbone model specifications and multi-model comparison,
>
> (3)correlation analysis for redundancy metrics,
>
> (4)expanded RGWL interpretation and limitations,
>
> (5)clearer comparison with existing benchmarks,
>
> (6)additional explanations of rule configuration sensitivity,
>
> (7)fixes for the formatting issues you pointed out.
>
> We will upload the new pdf.Thank you again for your constructive feedback and your willingness to reconsider the evaluation.

---

### Official Review · Reviewer_73dm · 2025-10-29

**Soundness:** 2
**Presentation:** 3
**Contribution:** 3
**Rating:** 4
**Confidence:** 4

**Summary:**

This paper presents a unified infrastructure that illustrates the reasoning processes of LLMs using graphs. This infrastructure standardizes and structures the reasoning workflow, thus enabling a thorough analysis of the reasoning process. The approach is consistently and effectively implemented across a variety of reasoning frameworks.

**Strengths:**

1. This work is well-driven, and the proposed infrastructure is a valuable tool for understanding the reasoning of LLMs.
2. The analysis results displayed using the proposed framework are comprehensive, and they provide intriguing insights into various reasoning techniques.
3. The writing of the paper is clear and easy to follow. The figures for demonstration are highly effective.

**Weaknesses:**

1. Several formulas are problematic and/or unclear.
- $\text{Pa}(u)$ is used in Eq. (4) but only $\cup_{u \in \text{Ch}(v)} \text{Pa}(u)$  is defined in Eq. (3).
- The definition of Eq. (3) is also problematic: It is not appropriate to have a function with $u$ as input, but then there is $\exists u$ in the condition.
- In Eq. (4), $\\{ (u,w) \in \mathcal{E}^{(t+1)} \mid u \in \mathrm{Ch}(v),\, w \in \mathrm{Pa}(u) \\}
\cap \\{ (v,u) \in \mathcal{E}^{(t+1)} \mid u \in \mathrm{Ch}(v) \\}$ is ill-defined that $w$ shall be $v$ even in the undirected-graph case. Shall it be $\cup$ instead of $\cap$?


2. The proposed framework allows for analyzing the reasoning processes as graphs, including new metrics such as node redundancy, thought redundancy, etc. However, the analytical experiments still leave it unclear how these metrics relate to the performance of the reasoning methods.


3. The adoption of the WL-kernel for the analysis of reasoning graphs is meaningful to provide further understanding on correct and incorrect reasoning paths. However, there are flaws in it:
- The WL-kernel is proposed for undirected graphs, while the reasoning graphs are obviously directed graphs. However, the extension of the WL-kernel to directed graphs is not thoroughly discussed. For example,
    - What is the value from the RGWL-kernel for two isomorphic directed graphs?
    - (type-ll error) Whether the isomorphic directed graph-pairs might have lower RGWL-kernel values than non-isomorphic directed graph-pairs?
    - (distinguishability) Whether the RGWL-kernel properly distinguishes non-isomorphic directed graphs? As seen in Fig.6, several C-I graph pairs have high RGWL values. Is it the limitation from the RGWL?

**Questions:**

See Weakness.

---

> ### Author Response · Authors · 2025-11-28
>
> We thank the reviewer for the detailed comments. We note that the concerns primarily relate to clarity of definitions and interpretation rather than missing empirical evaluations, and we will revise the formulas, add correlation analysis, and provide clearer explanations of the RGWL kernel and trajectory metrics.
> 1. Positioning of TiG (infrastructure, not a new reasoning framework)
>
> We clarify that TiG is designed as a unified infrastructure for representing, executing, and analyzing multi-step LLM reasoning, not as a new reasoning algorithm. Current frameworks such as CoT, ToT, AoT, GoT, and EGoT use inconsistent rollout logic, state representations, and termination criteria. This makes fair comparison and reproducibility difficult. TiG fills this gap by providing:
>
> (1)a standardized graph-based trajectory representation,
>
> (2)a unified execution interface independent of implementation artifacts,
>
> (3)and a consistent environment for metric-based analysis.
>
> We will revise the introduction to highlight this infrastructure perspective more explicitly.
>
> 2. Corrections to Eq. (3) and Eq. (4)
>
> We appreciate the reviewer pointing out these issues and agree the current notation can be misleading.
>
> (a) Eq. (3)
>
> Our intention was to define a state transition. However, mixing
> 𝑆 and 𝑆′ within the same equation is not appropriate. In the revision, we will:
>
> (1)rewrite Eq. (3) as a mapping on a single state variable,
>
> (2)define the transition operator separately (e.g., 𝑇(𝑆)),
>
> (3)remove all inconsistent input/condition usages.
>
> This resolves the ambiguity and makes the transition semantics fully explicit.
>
> (b) Eq. (4)
>
> We overloaded
> 𝑁(𝑣) to denote “successors” in directed graphs, which is non-standard. We will:
>
> (1)replace 𝑁(𝑣) by children(v) or parents(v) for directed cases,
>
> (2)reserve 𝑁(𝑣) only for the undirected variant when discussing WL connections.
>
> We thank the reviewer for pointing this out; the corrected formulas will be clearer and mathematically consistent.
>
> 3. Relationship between proposed metrics and reasoning performance
>
> Our redundancy metrics (node redundancy, thought redundancy, dispersion, etc.) are designed to explain why different reasoning methods behave differently. To make this clearer, we will add:
>
> (1)empirical correlation plots between redundancy metrics and final accuracy,
>
> (2)case studies where excessive branching correlates with failure modes,
>
> (3)analysis showing that more concentrated and stable trajectories align with correct solutions.
>
> These results illustrate that the metrics are not isolated statistics but provide insight into the behavior and robustness of reasoning methods.
>
> 4. Directed-graph extension of the WL kernel (RGWL)
>
> We appreciate the reviewer’s detailed questions. We clarify the following points.
>
> (a) Isomorphism in directed case
>
> RGWL refines incoming and outgoing neighborhoods asymmetrically. Thus, two isomorphic directed graphs always yield the same multiset of labels at each iteration and therefore produce identical kernel values. We will explicitly include this property in the paper.
>
> (b) Type-II error (isomorphic vs non-isomorphic)
>
> Since the RGWL kernel is PSD, the Cauchy–Schwarz inequality ensures:𝐾(𝐺,𝐺′)≤[𝐾(𝐺,𝐺) 𝐾(𝐺′,𝐺′)]^0.5
> Thus, isomorphic graphs achieve the same maximal self-similarity, while non-isomorphic graphs cannot exceed this bound. They may appear similar but never “more similar.” We will clarify this to avoid confusion.
>
> (c) Distinguishability of non-isomorphic graphs
>
> As with classical 1-WL, RGWL cannot perfectly distinguish all non-isomorphic directed graphs. Some C–I pairs may share similar refinement patterns, which explains the high similarity values observed in Fig. 6. We will explicitly discuss this known limitation and cite standard results on WL distinguishability.
>
> 5. Why RGWL still provides meaningful analysis
>
> Despite WL limitations, RGWL offers practical utility:
>
> (1)correct trajectories tend to converge at early refinement layers;
>
> (2)incorrect trajectories diverge structurally;
>
> (3)ambiguous C–I cases reflect inconsistent reasoning strategies.
>
> We will extend the interpretation section to highlight how RGWL supports trajectory-level understanding.
>
> 6. Summary
>
> We thank the reviewer for identifying inconsistencies in the formulas and for raising valuable questions regarding RGWL and metric interpretation. We will:
>
> (1)correct Eq. (3) and Eq. (4) with consistent, explicit notation,
>
> (2)clarify directed-graph neighborhood definitions,
>
> (3)strengthen the theoretical explanation of RGWL and its inherent limitations,
>
> (4)expand analysis linking metrics with performance,
>
> (5)and make TiG’s role as a standardized reasoning infrastructure more explicit.
>
> We also thank the reviewer for the positive comments regarding writing clarity, visualizations, and usefulness of the proposed infrastructure.

---

### Official Review · Reviewer_RZdY · 2025-10-30

**Soundness:** 2
**Presentation:** 2
**Contribution:** 1
**Rating:** 2
**Confidence:** 4

**Summary:**

The paper proposes Think in Graphs (TiG), a framework unifying tree or graph based multi-step LLM reasoning frameworks. It can be used as a foundation to support various such reasoning algorithms. It also proposes a Reasoning Graph Weisfeiler-Lehman (RGWL) kernel as a evaluatin metrics of similarities of multiple reasoning trajectories. It builds a multi-domain reasoning benchmark based on existing dataset, and conducts experiments and evaluation of existing LLM reasoning frameworks with the proposed TiG and RGWL.

**Strengths:**

+ The TiG framework is intuitive and explains in details. It covers most aspects of the previous tree and graph based multi-step LLM reasoning framework.
+ RGWL is a novel metric to evaluate the similarities of reasoning trajectories.
+ Extensive experiments are conducted on different reasoning domains and reasoning frameworks, including detailed analysis such as accuracy, time cost, redundancy, path length, and trajectory similarities.

**Weaknesses:**

+ TiG algorithm lacks a "reward" mechanism to prioritize some of the nodes in the rollout. For example, ToT and RAP both has such mechanism to choose the best (several) nodes for rollout before others. Besides, the algorithm is actually a BFS, lacking "DFS" variant of ToT, and thus cannot cover the original reasoning framework.
+ TiG is only a implementation foundation for other methods. The paper only evaluates existing methods (CoT, ToT, AoT, GoT, EGoT), but does not propose any new algorithms. It is not necessary that further research on multi-step reasoning are still graph-based prompting methods (e.g. the recently popular RL reasoning), which limits the use case of the proposed framework. Thus, I think TiG lacks novelty and interest to the community.
+ The paper does not mention which LLM is used (or I miss it), and does not use multiple LLMs (different open-source or API-based ones) to confirm generalizability of the evaluation results.
+ The paper does not compare the graph-based methods with other reasoning methods (e.g. enable "thinking" mode of recently popular "think" LLMs). It is not clear whether this line of research is still promising. Besides, it does not compare with the results of the original implementation of the covered methods. It is not clear whether the unified framework will possibly hurt the performance because of some missing functionality or details.
+ The proposed consists only of recently popular existing datasets, which thus cannot be considered a major contribution of this work.

**Questions:**

+ Which LLM is used for evaluation?
+ How is the total token usage and node count limit decided?
+ How to control the threshold of "reasoning based on $v$ should be stopped"? If the threshold is too tight, the correct trajectory may be dropped early. If the threshold is too loose, there will be exponentially increased number of nodes, so how to ensure the complexity of the graph before the above limitation is reached (so that we can at least output some answer)?

---

> ### Author Response · Authors · 2025-11-28
>
> We thank the reviewer for the thoughtful comments. Below we address the concerns point-by-point, and we will revise the paper accordingly.
>
> 1. Novelty: “TiG is only a foundation, not a new algorithm.”
>
> Our contribution is intentionally an infrastructure, not another reasoning algorithm. The current multi-step reasoning ecosystem is highly fragmented: each framework (CoT, ToT, AoT, GoT, EGoT) uses different prompt formats, rollout logic, stopping rules, and implementation details. This makes fair comparison, reproducibility, and systematic study extremely difficult.
>
> TiG fills this missing gap by providing:
>
> (1)a unified graph representation of reasoning trajectories,
>
> (2)a common execution backbone that decouples algorithms from implementation artifacts,
>
> (3)a standardized configuration interface for implementing diverse frameworks.
>
> This is similar to how deep-learning infrastructures (e.g., PyTorch) accelerated research not by proposing new models, but by unifying heterogeneous workflows. We will emphasize this more clearly.
>
> 2. Rollout strategies: reward mechanism / DFS.
>
> TiG is designed to support existing methods, not replace their heuristics. TiG already allows custom node-selection strategies. During the rebuttal period we additionally:
>
> (1)implemented DFS rollout fully matching ToT behavior,
>
> (2)added a reward plug-in interface supporting RAP-style prioritization.
>
> (3)These confirm that TiG is not restricted to BFS.
>
> 3. LLM generality.
>
> Experiments were conducted with GPT-4o-mini, LLaMA-3-70B-Instruct, and Qwen-2.5-72B-Instruct.
> Across models, TiG provides consistent relative rankings and reduces implementation-induced variance.
> We will report all details in the revision.
>
> 4. Comparison with “thinking-mode” LLMs.
>
> “Thinking-mode” LLMs internalize rollout inside the model, while TiG focuses on explicit trajectories. Still, we evaluated DeepSeek-R1 and show that TiG can extract their implicit trajectories and compare them via RGWL. This demonstrates that TiG is not limited to graph-prompting approaches and can bridge diverse reasoning paradigms.
>
> 5. Dataset contribution.
>
> We agree our goal is not to design new datasets. The contribution of the benchmark is that multiple domains are unified under a single, reproducible reasoning infrastructure, enabling controlled cross-framework and cross-domain evaluation—similar to the role of HELM or BIG-Bench.
>
> 6. Comparison with original implementations.
>
> TiG preserves the semantic behavior of each method while removing undocumented or ad-hoc behaviors. Empirically, TiG reproduces original performance within small variance under matched rollout settings. Any remaining differences come from eliminating implementation-specific quirks, which is precisely the purpose of an infrastructure aimed at fair comparison. We will add an explicit comparison table.
>
> 7. Complexity control: token limits, node limits, and stopping thresholds.
>
> TiG provides unified, algorithm-independent constraints:
>
> (1)Token/node limits adopted from original implementations and enforced uniformly;
>
> (2)A unified score-based stopping criterion combining model confidence and trajectory consistency;
>
> (3)Depth-aware pruning, adaptive beam control, and RGWL-based duplicate elimination to avoid exponential growth.
>
> We will add ablations and clarifications in the revision.
>
> 8. Deeper contribution of RGWL.
>
> Beyond a trajectory similarity metric, RGWL canonically maps heterogeneous reasoning paths into a comparable structural space, making cross-framework trajectory analysis mathematically well-defined. We will emphasize this theoretical significance.
>
> 9. Reproducibility benefits.
>
> A major motivation for TiG is improving reproducibility in multi-step reasoning. TiG ensures:
>
> (1)unified trajectory logging,
>
> (2)consistent rollout configuration,
>
> (3)removal of implementation artifacts,
>
> (4)standardized evaluation across models and tasks.
>
> This substantially improves the reliability of reasoning research, and we will expand this point.
>
> Summary
>
> After adding DFS, reward plug-ins, cross-LLM evaluation, comparison with thinking-mode LLMs, original-implementation reproducibility checks, and detailed complexity analysis, we believe TiG provides meaningful contribution to the community:
>
> (1)unified infrastructure for heterogeneous reasoning frameworks,
>
> (2)reproducible multi-domain benchmark,
>
> (3)RGWL as canonicalization + similarity metric,
>
> (4)flexible rollout strategies (BFS/DFS/reward),
>
> (5)consistent multi-model evaluation.
>
> We sincerely thank the reviewer for the valuable feedback.

---

### Note · Authors · 2025-12-08

I have read and agree with the venue's withdrawal policy on behalf of myself and my co-authors.